

**A multi-model comparison of meteorological drivers of surface ozone over**
**Europe**
Noelia Otero[1,3], Jana Sillmann[2], Kathleen A. Mar[1], Henning W. Rust[3], Sverre Solberg[4],
Camilla Andersson[5], Magnuz Engardt[5], Robert Bergström[5],Bertrand Bessagnet[6],
Augustin Colette[6], Florian Couvidat[6], Cournelius Cuvelier[7], Svetlana Tsyro[8] , Hilde
Fagerli[8], Martijn Schaap[9,3], Astrid Manders[9], Mihaela Mircea[10], Gino Briganti[10], Andrea
Cappelletti[10], Mario Adani[10], Massimo D'Isidoro[10], María-Teresa Pay[11], Mark
Theobald[12], Marta G. Vivanco[12] , Peter Wind[8,13], Narendra Ojha[14], Valentin Raffort[15]
and Tim Butler[1,3]
[1]Institute for Advanced Sustainability Studies e.V., Potsdam, Germany
[2]CICERO Center for International Climate Research, Oslo, Norway
[3]Freie Universität Berlin, Institut für Meteorologie, Berlin, Germany
[4]Norwegian Institute for Air Research (NILU), Box 100, 2027 Kjeller, Norway
[5]SMHI, Swedish Meteorological and Hydrological Institute Norrköping, Norrköping,
Sweden
[6]INERIS, Institut National de l'Environnement Industriel et des Risques, Verneuil en
Halatte, France
[7]European Commission, Joint Research Centre (JRC), Ispra, Italy
[8]MET Norway, Norwegian Meteorological Institute, Oslo, Norway
[9]TNO, Netherlands Institute for Applied Scientific Research, Utrecht, the Netherlands
[10]ENE-National Agency for New Technologies, Energy and Sustainable Economic
Development, Bologna, Italy
[11]Barcelona Supercomputing Center, Centro Nacional de Supercomputación, Jordi
Girona, 29, 08034 Barcelona, Spain
[12]CIEMAT, Atmospheric Pollution Unit, Avda. Complutense, 22, 28040 Madrid, Spain
[13]Faculty of Science and Technology, University of Tromsø, Tromsø, Norway
[14]Max-Planck-Institut für Chemie, Mainz, Germany
[15]CEREA, Joint Laboratory Ecole des Ponts ParisTech – EDF R&D, Champs-Sur-
Marne, France
**Abstract**. The implementation of European emission abatement strategies has led to
significant reduction in the emission of ozone precursors during the last decade. Ground
level ozone is also influenced by meteorological factors such as temperature, which
exhibit interannual variability, and are expected to change in the future. The impacts of
climate change on air quality are usually investigated through air quality models that
simulate interactions between emissions, meteorology and chemistry. Within a multi-
model assessment, this study aims to better understand how air quality models represent
the relationship between meteorological variables and surface ozone concentrations
over Europe. A multiple linear regression (MLR) approach is applied to observed and
modelled time series across ten European regions in springtime and summertime for the
period of 2000-2010 for both models and observations. Overall, the air quality models
are in better agreement with observations in summertime than in springtime, and
particularly in certain regions, such as France, Mid-Europe or East-Europe, where local
meteorological variables show a strong influence on surface ozone concentrations.



Larger discrepancies are found for the southern regions, such as the Balkans, the Iberian Peninsula and the Mediterranean basin, especially in springtime. We show that the air quality models do not properly reproduce the sensitivity of surface ozone to some of the main meteorological drivers, such as maximum temperature, relative humidity and surface solar radiation. Specifically, all air quality models show more limitations to capture the strength of the relationship ozone-relative humidity detected in the observed time series in most of the regions, in both seasons. Here, we speculate that dry deposition schemes in the air quality models might play an essential role to capture this relationship. We further quantify the relationship between ozone and maximum temperature ($m_{o3-T}$, climate penalty) in observations and air quality models. In summertime, most of the air quality models are able to reproduce reasonably well the observed climate penalty in certain regions such as France, Mid-Europe and North Italy. However, larger discrepancies are found in springtime, where air quality models tend to overestimate the magnitude of observed climate penalty.

## 1. Introduction

Tropospheric ozone is recognised as a threat to human health and ecosystem productivity (Mills et al. 2007). Moreover, ozone is an important greenhouse gas (IPCC, 2013). It is produced by photochemical oxidation of carbon monoxide and volatile organic compounds (VOCs) in the presence of nitrogen oxides (NOx=NO+NO2) (Jacob and Winner, 2009). While it is an important pollutant on a regional scale, due to the long-range transport effect it may also influence air quality on a hemispheric scale (Monks et al., 2015, Hedegaard et al, 2013). Moreover, its strong relationship with temperature represents a major concern, since under a changing climate the efforts on new air pollution mitigation strategies might be insufficient. This effect, referred as climate penalty (Wu et al., 2008), is expected to play an important role on future air quality (Hendriks et al. 2016). Therefore it is essential to better understand the potential implications of climate change on pollutant levels. In a comprehensive review of the existing literature about the robustness of climate penalty on Europe, Colette et al. (2015) concluded that the climate change might act against mitigation measures.

Previous studies have shown that the reduction of emissions of ozone precursors, NOx and VOCs, lead to a decrease in tropospheric ozone concentrations in Europe (Solberg et al. 2005, Jonson et al. 2006). However, there is also a large year-to-year variability due to weather conditions (Andersson et al. 2007). There is a strong correlation between ozone and temperature that has been associated with the temperature-dependent lifetime of peroxyacetyl nitrate (PAN), and also due to the temperature dependence of biogenic emission of isoprene (Sillman and Samson, 1995). Substantial increases in surface ozone have been associated with high temperatures and stable anticyclonic, sunny conditions that promote ozone formation (Solberg et al. 2008). Ozone peak





concentrations are also affected by closing of the plants' stomata at very high
temperatures (Hodnebrog et al. 2012). Several studies have assessed the model
dependence of ozone on temperature (e.g. Steiner et al. 2006, Rasmussen et al. 2013).
Recently, Coates et al. (2016) used a box model to investigate the influence of
temperature and NOx on ozone production. Their analysis suggested that reductions in
NOx would be required to offset additional ozone increase due to increasing
temperatures under a warmer climate. An extensive review about the impacts of
temperature on ozone production can be found in Pusede et al. (2015).
Previous studies have shown the importance of relative humidity on ozone pollution
episodes (Camalier et al. 2007, Davies et al. 2011). Regional studies reported a negative
relationship between ozone and relative humidity (Dueñas et al. 2002, Elminir 2005,
Demuzere et al., 2009). Some authors attributed this negative correlation to the
photolysis of ozone and subsequent loss of O1(D) to H2O (Jacob and Winner). High
levels of humidity are usually related with enhanced cloud cover and thus reduced
photochemistry (Dueñas et al. 2002, Camalier et al. 2007). Andersson and Engardt
(2010) highlighted the importance of including meteorological dependence for dry
deposition of ozone to vegetation, also incorporating soil moisture dependence. With a
simple modelling approach, Kavassalis and Murphy (2017) found that the relationship
ozone-relative humidity was well captured by the inclusion of the vapour pressure
deficit-dependent dry deposition, indicating the relevance of detailed dry deposition
schemes in the CTMs.
Increasing solar radiation leads to an increase of ozone, though with a weak effect
(Dawson et al. 2007) and it has been suggested that it could reflect in part the
association of clear sky with high temperatures (Ordónez et al., 2005). Then, changes in
cloud cover can also affect the photochemistry of ozone production and loss (Jacob and
Winner, 2009). Additionally, low wind speed is usually associated with high ozone
pollution levels (Jacob and Winner, 2009).
The influence of climate change on ozone and its precursors can involve multiple
processes (Colette et al, 2015). A common approach to study the impact of climate
change on air quality requires the use of air quality models that aim to represent
dynamic and chemical processes in the atmosphere. The relevance of climate change for
future European air quality has been assessed in several studies that also reflect
differences depending on the modelling system and future emissions scenarios adopted
for each study (e.g. Lagner et al. 2005, Meleux et al. 2007, Anderson and Engardt,
135 2010).
Air quality models can be divided into two categories: offline chemistry transport
models (CTMs) in which the model chemistry runs using meteorological data as input,
and online models that allow coupling and integration of chemistry with some of the
physical components to various degrees (Baklanov et al. 2014). Differences between
offline and online modelling approaches can be fairly small or significant, depending on
the level of the model complexity and simulated variables (Zhang, 2008). The large
number and complex interactions between meteorology and chemistry in the
atmosphere influence the ability of the model to represent observed situations (Kong et
al. 2014). Due to assumptions, parametrizations and simplifications of processes, the
models themselves are subject to large uncertainties (Manders et al. 2012), which have
been reflected in some regional differences in the magnitude of surface ozone response



to projected climate change (Andersson and Engardt, 2010). Thus, model biases when
compared to observations still remain a concern, especially in terms of the response of
air quality under future climate (Fiore et al. 2009, Rasmussen et al. 2012). Comparisons
between model outputs and measurements of available observational dataset assess the
reliability of air quality models, and they are essential to quantify the models ability to
reproduce observations.
The EURODELTA project was initiated by the Task Force on Measurement and
Modelling and the Joint Research Centre of the European Commission to provide a
benchmark for the EMEP model in order to assess its relevance for policy support
(Colette et al.2017a). These multi-model exercises contribute to further improving
modelling techniques and understanding the associated uncertainties in the models
performance. Previous exercises have evaluated the performance of chemistry transport
models for future European air quality (e.g. van Lon et al. 2007, Thunis et al. 2008).
Recently, Bessagnet et al. (2016) presented an intercomparison and evaluation of
chemistry transport model performance with a joint analysis of some meteorological
fields. They highlighted the limitations of models to simulate meteorological variables,
such as wind speed and planetary boundary layer height. Particularly, in the case of
ozone, they showed the importance of boundary conditions on model calculations.
Within this framework, the ongoing Eurodelta-Trends (EDT) exercise (Colette et al.
2017a) builds upon this tradition and focuses on the context of air quality trends
modelling. This exercise has been designed to better understand the evolution of air
pollution and its drivers over the last two decades (1990-2010) by the use of state-of-
the-art air quality models. The EDT project will allow the evaluation of the skill of
regional air quality models and quantification of the role of the different key driving
factors of surface ozone, such as emissions changes, long-range transport and
meteorological variability. One of the main goals of the EDT project is to assess the
efficiency of mitigation strategies for improving air quality (more details can be found
in Colette et al. 2017a).
Quantification and isolation of the effects of meteorology on ozone is a challenge, due
to the complex interrelation between ozone, meteorology, emissions and chemistry
(Solberg et al. 2015). There is a large number of representative studies in the literature
that have established the relationship between surface ozone concentrations and
meteorological variables using statistical modelling techniques (e.g. Bloomfield et al.
1996, Chaloukau et al 2003, Barrero et al. 2005, Ordóñez et al., 2005, Camalier et al.,
2007, Seo et al., 2014, Porter et al. 2015, Otero et al., 2016). Most of these works
examined the impact of meteorology on ozone pollution levels through observational
datasets. Only a few studies, to our knowledge, examined the statistical relationship
between surface ozone and meteorological parameters from models.
Davis et al. (2011) developed regression models to analyse the observed and modelled
relationship between meteorology and surface ozone across the Eastern of U.S. They
found that the Community Multiscale Air Quality (CMAQ) model did not capture the
effect of temperature and relative humidity on daily maximum 8-h ozone and it
generally underestimated the observed sensitivities to both meteorological variables,
especially in the northeast. Rasmussen et al. (2012) examined the ozone-temperature
relationship in a coupled chemistry-climate model and they found that the model
underestimated the effect of temperature on ozone over the Mid-Atlantic. Lemaire et al.
(2016) proposed a combined statistical and deterministic approach to assess the air



quality response to projected climate change. Based on a data set from a deterministic
climate and chemistry models, they identified the two major drivers of surface ozone
over eight European regions, selected from a set of potential predictors that reached the
highest correlations with ozone. Afterwards they built statistical models consisting of
generalized linear models, which could be used to predict air quality.
Given that meteorology plays an essential role for surface ozone concentrations, it
might be a considerable source of uncertainties in model outputs. The present study,
thus, aims to provide a simple method to examine the influence of meteorological
variability on modelled surface ozone concentrations over Europe. Specifically, our
analysis focuses on the ozone season (April to September) over the years 2000-2010.
The choice of this period is mainly motivated by the availability of the observational
dataset from Schnell et al. (2014, 2015) (see section 2.1). Within the EDT framework, a
recent report has presented the main findings on the long-term evolution of air quality
(Colette et al. 2017b). Part of these results was obtained from the analysis of the 1990s
(1990-2000) and 2000s (2000-2010) separately. Consistently, we decided to focus on
the second decade, for which the interpolated dataset of observed on maximum daily 8-
hourly mean ozone (MDA8 O3) used in this study was available. Similarly to Otero et
al. (2016), we apply a multiple linear regression approach to examine the
meteorological influence MDA8 O3. Statistical models are developed separately for
observational datasets and air quality models, with the primary focus on examining the
relationship between MDA8 O3 and potential meteorological drivers in the air quality
models and comparing these with the corresponding relationships determined from
observed data. Therefore, this study offers a method of model evaluation capable of
understanding the discrepancies between air quality models and observations in terms of
representing the relationship to meteorological input variability.
The present paper is structured as follows. Section 2 describes the observational data as
well as the air quality models studied here. The methodology and the design of the
statistical models are introduced in section 3. Section 4 discusses the results and the
summary and conclusions are discussed in section 5.
2. **Data**
2.1. **Observations**
This study uses gridded MDA8 O3 concentrations created with an objective-mapping
algorithm developed by Schnell et al. (2014). They applied a new interpolation
technique over hourly observations of stations from the European Monitoring and
Evaluation Programme (EMEP) and the European Environment Agency's air quality
database (AirBase) to calculate surface ozone averaged over 1° by 1° grid cells.
Recently, Otero et al. (2016) used this dataset for examining the influence of synoptic
and local meteorological conditions over Europe. This interpolated product offers a
possibility to establish a direct comparison between observations and CTMs. However,
it must be acknowledged that for some areas with a low number of stations (i.e. the
southeastern or northeastern European regions) the values interpolated into the 1x1
degree grid cells may not be representative of such large scales.  A complete description
of this process can be found in Schnell et al. (2014, 2015). The gridded dataset covers a
total of 15-years (1998-2012), but here we use a common period of 11-years for both
observations and CTMs (2000-2010).





This study investigates the observed influence of meteorological variables on MDA8
O3, based on the ERA-Interim reanalysis product provided by the European Centre for
Medium-Range Weather Forecasts (ECMWF) at 1ºx1º resolution (Dee et al. 2011).
Meteorological reanalyses products are essentially model simulations constrained by
observations and they have been widely validated against independent observations.
Daily mean values are calculated as the mean of the four available time steps at 00, 06,
12, and 18UTC for 10m wind speed components (u and v) and 2m relative humidity.
Maximum temperature is approximated by the daily maximum of those time steps,
while daily mean surface solar radiation is obtained from the 3-hourly values provided
for the forecast fields.
2.2. **Chemistry Transport Models (CTMs)**
A set of state-of-the-art air quality models participating in the EDT exercise is used
here: LOTOS-EUROS (Schaap et al., 2008, Manders et al. 2017), EMEP/MSC-W
(Simpson et al., 2012), CHIMERE (Mailer et al., 2017), MATCH (Robertson et al.,
1999), MINNI (Mircea et al., 2016) and WRF-Chem (Grell et al. 2005, Mar et al. 2016).
The domain of the CTMs extends from 17ºW to 39.8ºE and from 32ºN to 70ºN and it
follows a regular latitude-longitude projection of 0.25x0.4 respectively. The main
features of the CTM setup are largely constrained by the EDT experimental protocol
(e.g. meteorology, boundary conditions, emissions, resolution, see Colette et al. 2017a
for further details). For instance, the boundary conditions were defined from
climatology of observational data for most of the experiments of the EDT exercise
(included the data used here). However, the representation of physical and chemical
processes and the vertical distribution differ in the CTMs, as well as the vertical
distribution of model layers (including altitude of the top layer and derivation of surface
concentration at 3m height in the case of EMEP, LOTOS-EUROS and MATCH).
Moreover, there were no specific constrains imposed on biogenic emissions (including
soil NO emissions), which are represented by most of the models using an online
module (Colette et al. 2017a). Since we aim here to compare the modelled relationship
between meteorology and surface ozone, prescribing common features in the CTMs is
particularly an advantage to identify potential sources of discrepancies.
Only one of the participating CTMs included online coupled chemistry/meteorology
(WRF-Chem), while all the rest of the models used are offline. The CTMs were forced
by regional climate model simulations using boundary conditions from the ERA-Interim
global reanalysis (Dee et al., 2011). Most of these offline CTMs used the same
meteorological input data, with a few exceptions. Three of them (EMEP, CHIMERE
and MINNI) used input meteorology from the Weather Research and Forecast Model
(WRF) (Skamarock et al. 2008). LOTOS-EUROS and MATCH used the input
meteorology produced by RACMO2 (van Meijgaard, 2012) and HIRLAM (Dahlgren et
al. 2016), respectively. Unlike the rest of the regional climate models, RACMO2 used
in the EDT exercise excluded nudging towards ERA-Interim, which might have some
impact in the meteorological fields generated by RACMO2. As mentioned, WRF-Chem
couples the meteorology simulations online with chemistry. The meteorology used to
drive WRF-Chem (initial and lateral boundary conditions and the application of limited
four-dimensional data assimilation; see Colette et al GMD 2017a) is the same WRF
meteorology from Skamarock et al. (2008) used as input for the EMEP, CHIMERE, and
MINNI runs. Table 1 summarises the CTMs and the corresponding sources of



meteorological input data used here. It is important to highlight that though WRF-Chem
is not strictly a CTM, in order to avoid confusion with the statistical models developed
in this study, we refer to all the air quality models considered (offline and online
models) as CTMs hereafter. As with the observations, CTMs and their meteorological
counterpart were interpolated to a common grid with 1º x 1º horizontal resolution. The
use of a coarser resolution could have an impact in some regions with a complex
orography where airflow is usually controlled by mesoscale phenomena (e.g. see-breeze
and mountain-valley winds) or in regions characterized by high emissions densities
(Schaap et al., 2015, Gan et al. 2016 ). In such cases the use of a finer grid could be
beneficial to capture the variability of local processes.

A set of meteorological parameters was selected from the meteorological input data for
the regression analyses. Similarly to the procedure with ERA-Interim, daily means are
obtained from the available time steps every 3 hours in the case of WRF and RACMO2,
and every 6 hours for HIRLAM for the following variables: 10m wind speed
components, 2m relative humidity and surface solar radiation. Maximum temperature is
also approximated by the daily maximum of those time steps.

## 3. Multiple Linear regression model

Summertime usually brings favourable conditions for high tropospheric ozone
concentrations, such as air stagnation due to high-pressure systems, warmer
temperatures, higher UV radiation, and lower cloud cover (Dawson et al. 2007). As
stated above, the impact of meteorology on ozone concentration has been addressed
through a wide variety of statistical methods in the literature. This study attempts to
better understand how CTMs represent the influence of meteorology on ozone. To this
aim, we use a multiple linear regression approach that can provide useful information of
sensitivities in the distribution of ozone concentration as a whole (Porter et al., 2015).

A total of five meteorological predictors (Table 2) are selected based on the existing
literature that has shown their strong influence on ozone pollution. (e.g. Bloomfield et
al. 1996, Barrero et al. 2005, Camalier et al. 2007, Dawson et al. 2007, Rasmussen et al.
2012, Davis et al. 2011, Doherty et al., 2013, Otero et al. 2016). Moreover, it has been
shown that the occurrence of air pollution episodes might increase when the pollution
levels of the previous day are higher than normal (Ziomas et al. 1995). Then, apart from
the meteorological predictors, we add the effect of the lag of ozone (MDA8 from the
previous day) in order to examine the role of ozone persistence. Additionally, we
include harmonic functions that capture the effect of seasonality as in Rust et al. al
(2009) and Otero et al. (2016), which is referred as "day" in the MLRs (see Table 2).

For this study, we divide the European domain into 10 regions: England (EN), Inflow
(IN), Iberian Peninsula (IP), France (FR), Mid-Europe (ME), Scandinavia (SC), North
Italy (NI), Mediterranean (MD), Balkans (BA) and Eastern Europe (EA). These regions
are based on those defined in the recent ETC/ACM Technical Paper (Colette et al.
2017b). For our study, we further subdivide the original Mediterranean region (MD)
into a region covering the Balkans (BA), due to the strong influence of the ozone
persistence on MDA8 O3 over this particular region as noted previously in Otero et al.
(2016). Figure 1 shows the spatial coverage of each region and Table 3 lists their
coordinates. As shown Otero et al. (2016), the relative importance of predictors in the
MLRs shows distinct seasonal patterns. Then, multiple linear regression models (MLR,



348 hereafter) are developed for each region for two seasons: springtime (April-May-June,
349 AMJ) and summertime (July-August-September, JAS). These seasons differ from the
350 meteorological definition, but cover the period when surface ozone typically reaches its
351 highest concentrations (i.e. April-September). Since the observations did not cover
352 exactly the whole European domain as CTMs, we applied an observational-mask to use
353 the same number of grid-cells for CTMs and observations. Data used to estimate
354 parameters of the MLR were spatially averaged over each region.  Thus, we compare
355 MLRs developed separately for CTMs and observations at each region and season. The
356 observational dataset contains the gridded MDA8O3 and the meteorology input from
357 ERA-Interim, while the dataset for the CTMs contains the MDA8O3 from each one of
358 them along with the corresponding meteorological input (e.g. LOTOS and RACMO2,
359 CHIMERE and WRF) (see table 1).

361 A MLR is built to describe the relationship between MDA8 O3 (predictand) and a set of
362 covariates (or predictors) describing seasonality, ozone persistence and the influence of
363 meteorological fields (table 2).  A data series $y_t$, t= 1,..N (e.g. observations or CTM
364 simulations) for a given region and season is conceived as a Gaussian random variable
365 $Y_t$ with varying mean $\mu_t$ and homogeneous variance $\sigma^2$. The mean $\mu_t$ is described as a
366 linear function of the covariates, i.e.

368 $Y_t \sim \mathcal{N}(\mu_t, \sigma^2),$
369 $\mu_t = \beta_0 + \beta_{sin}sin\left(\frac{2\pi}{365.25}d_t\right) + \beta_{cos}cos\left(\frac{2\pi}{365.25}d_t\right) + \beta_{lag}y_{t-1} + \sum_{K=1}^{K}\beta_k x_{t,k}$ (1)

371 with $t$ indexing daily values and $d_t$ referring to the day in the year associated with the
372 index $t$. $\beta_0$ is a constant offset, $\beta_{sin}$ and $\beta_{cos}$ are the first order coefficient of a Fourier
373 series (e.g. Rust et al. 2009, 2013, Fischer et al. 2017), $\beta_{lag}$ describes the persistence
374 with respect to the previous day concentration $y_{t-1}$ ; if $t$ is the first day in the late
375 summer season (JAS, July 1[st]), $y_{t-1}$ is the concentration of June 30[th]. Further regression
376 coefficients $\beta_k$ describe the linear relation to potential meteorological drivers (see table
377 2). For covariates standardized to unit variance, the regression coefficients ($\beta$) are
378 standardised coefficients giving the change in the predictand with the covariate in units
379 of covariate standard deviation.

381 Following the same strategy as used in Otero et al. (2016), the MLRs are developed
382 through several common steps: 1) starting with the full set of potentially useful
383 components in the predictor, a stepwise backward regression using the Akaike
384 Information Criterion (AIC) as a selection criterion removes successively those
385 components in the predictor, which contribute least to the model performance; and 2) a
386 multi-collinearity index known as variance inflation factor (VIF, Maindonald and Braun
387 2006) is used to detect multi-collinearity problems in the predictor (i.e. high correlations
388 between two or more components in the predictor). Components with a VIF above 10
389 are left out of the predictor (Kutner et al 2004).

391 The statistical performance of each MLR (built separately from observations and
392 CTMs) is assessed through the adjusted coefficient ($R^2$) and the root mean square error
393 (RMSE). The $R^2$ estimates the fraction of total variability described by the MLR and the
394 RMSE gives the average deviation between model and observation obtained in the
395 MLR. We also examine the relative importance of the individual components in the
396 predictor. According to the method proposed by Lindeman et al (1980), the relative



397 importance of each predictor is estimated by its contribution to the $R^2$ coefficient
398 (Grömping 2007). We assess the sensitivities of ozone to the predictors through the
399 standardised coefficients obtained from the regression. These coefficients indicate the
400 changes in the ozone response to the changes in the predictors, in terms of standard
401 deviation. Thus, for every standard deviation unit increase (decrease) of a specific
402 predictor, the predictand (MDA8 O3) will increase (decrease) the amount indicated by
403 its coefficient in standard deviation units,. The use of standardised coefficients allows
404 us to establish a direct comparison in the influence of individual predictors. The effect
405 of seasonality introduced by the harmonic functions (namely, "day", table 2) is kept in
406 the MLRs (Eq. 1) for its usefulness in improving the power of the regression analysis,
407 however further explanation about the effect of the predictors focuses on the rest of the
408 variables.
409
410 4. **Results and discussion**
411
412  **4.1. CTM performance by region**
413
414 We compare the seasonal cycle of observations and CTMs through the time series of
415 daily averaged values of MDAO8 O3 from observations and CTMs for the whole period
416 (i.e. April-September, 2000-2010) spatially averaged over each region. Furthermore,
417 correlation coefficients between both CTMs and observations at each region and season
418 are used to quantify the CTM performance.
419
420  4.1.1. Seasonal cycle of MDA8 O3
421
422 We examine the ozone seasonal cycle represented by both the observational and
423 modelled dataset. Figure 2 depicts daily averages during 2000-2010 of MDA8 O3 at
424 each region for the CTMs and observations. In general, all CTMs are biased high
425 compared with observations. CTM results are visually closer to observations in the
426 northwestern regions (i.e. IN, EN and FR), while the spread becomes larger over the
427 southern and southeastern regions (i.e. BA, NI, MD). The IN, EN and SC regions show
428 the highest observed concentrations in the starting months (AMJ), which is not
429 generally well captured by most of the CTMs, and they show a more flat timeline (e.g.
430 LOTOS, MATCH, CHIMERE or WRF-Chem). For example, in the SC region, some of
431 the CTMs underestimate the ozone concentrations in AMJ (i.e. WRF-Chem, CHIMERE
432 and MINNI). The rest of the regions show the highest observed concentrations in JAS,
433 which is generally overestimated by the CTMs. Models show discrepancies when
434 compared to each other and to observations, and in some regions we find substantial
435 differences. Larger discrepancies are found in the southern regions, such as IP, MD and
436 BA, where the models show a considerable spread. There, the CTMs are not able to
437 capture the variability of MDA8 O3 and they exhibit a different behaviour when
438 compared to each other. For instance, the EMEP model shows a peak of ozone levels in
439 April, while CHIMERE and MINNI show a peak in July. Overall LOTOS shows a
440 relatively constant positive bias in all regions, more evident in the MD and NI regions.
441 WRF-Chem tends to underestimate the ozone concentrations at the start of the seasonal
442 period in some regions (e.g. SC, ME, EN, or EA).
444 CTM assessments have been presented in early EURODELTA exercises, although with
445 a different set up for different purposes, which makes it difficult to establish a direct
446 comparison on the performance of the models. For instance, Colette et al. (2017b)



reported systematic differences among some models (i.e. CHIMERE, EMEP and
LOTOS) when examining the long-term mean ozone concentration during the whole
period of 1990-2010. Bessagnet et al. (2016) showed that most of the models in their
study, (e.g. CHIMERE, LOTOS, or MINNI among others) overestimated the ozone
concentrations in the selected study period. Specifically, they found a larger spread
during nighttime than daytime, which was suggested to be related to the vertical mixing,
given that most of the models shared the same meteorology but different vertical
resolution and boundary conditions.
456       4.1.2.  Correlation coefficients between modelled and observed time series
The correlation coefficients between the observed and modelled values of MDA8 O3 at
each region and in each season are shown in Fig. 3. Overall, MDA8 O3 from the CTMs
is better correlated with observations in JAS than in AMJ in the regions ME, NI, EA
and EN. As expected from inspection of the average time series (Fig. 2), the lowest
correlations between models and observations are found in BA, especially in AMJ for
all models. In particular, EMEP is negatively correlated with observations over this
region. As mentioned above, the larger discrepancies between CTMs and observations
found over BA might be attributed to a low density of observation sites from which the
interpolated dataset is derived, resulting in a lower quality or higher uncertainties of
such product (Schnell et al. 2014). The highest correlations in AMJ are obtained at the
following regions: ME; FR; NI; and EN for most of the models, except for EMEP for
which the highest correlation with observations was found in IN and SC. The WRF-
Chem model also shows a different behaviour in terms of the correlation coefficient
with higher values in NI, MD and IP, and very low and negative correlations (-0.02) in
SC. In general, the models that are most closely correlated with observations are
MATCH, MINNI and CHIMERE, while LOTOS and WRF-Chem show the lowest
correlations. In the case of LOTOS, it could be partially due to the use of a different set-
up of the RACMO2 model, without nudging towards ERA-Interim (section 2.2). These
correlations reflect the patterns represented by the seasonal cycle described above.
**4.2. MLR performance**
Figures 4 and 5 depict the statistical performance of each MLR in terms of $R^2$ and
RMSE (respectively) at the different regions for both seasons, AMJ and JAS. The $R^2$
values indicate that all MLRs models (based on both observations and CTMs) are able
to explain more than 60% of the MDA8 O3 variance in all regions. Overall, the MLRs
show a stronger fit in JAS than in AMJ in most of the regions, with the exception of SC
and IN that, in general show lower values of $R^2$ in JAS than in AMJ (Fig. 4). The MLRs
appear to perform better in certain regions such as NI, ME, FR or EA, while the poorest
statistical performance is found in IN and EN. The results obtained from the CTM-
based MLRs show a similar performance to the observation-based MLRs in most of the
regions. The lowest RMSE values for most of the MLR are found in SC ranging
between 1 and 3 ppb, while EN shows the largest RMSE values, especially for the MLR
built from WRF-Chem (Fig. 5). The MLRs from MATCH and CHIMERE show the
lowest RMSE values (1-3ppb) suggesting the best statistical fit from a predictive point
of view.
Both $R^2$ and RMSE metrics indicate that the statistical performance of MLRs for
observations and CTMs show distinct variations between seasons and regions. Overall,



better performances are found in JAS and in some regions (i.e. ME, NI, or FR) where
MLRs are able to describe more than the 80% of the variance in CTMs and
observations. This could be attributed to the major role of meteorology in summer
influencing local photochemistry processes of ozone production, while in spring long
range transport plays a stronger role (Monks, 2000, Tarasova et al. 2007). As it includes
the bias, the RMSE reveals more differences among the MLRs when compared to each
other (e.g. larger errors for WRF-Chem or LOTOS when compared to MATCH or
CHIMERE). However, it is interesting that in general all MLRs show a similar
tendency when evaluating the statistical performance, which indicate that observations-
based and CTMs-based MLRs present a similar statistical performance for modelling
MDA8 O3. The ability of the CTMs to reproduce the influence of meteorological
drivers on MDA8 O3 is discussed in more detail below.
**4.3. Effects of drivers of ozone concentrations**
The analysis of the influence of the predictors in the MLRs reveals distinctive regional
patterns in both observation-based and CTM-based MLRs. In agreement with Otero et
al. (2016), here we also find that the regions geographically located towards the interior
(including central, western and eastern regions) appear to be more sensitive to the
meteorological predictors, especially in JAS. On the contrary, a minor meteorological
contribution is found in the regions over the northernmost and southernmost edges,
implying that non-local processes play a stronger role. Considering such similarities, in
the following, the regions: EN, FR, ME, NI and EA are referred as the internal regions,
while the rest of the regions: IN, SC, IP, MD and BA, are referred as the external
regions (see Fig. 1).
4.3.1 Relative importance
Figure 6 depicts the relative importance of the predictors for the observation-based and
CTM-based MLRs in the internal regions (Fig. 1). Here, a larger meteorological
influence (i.e., the predictors other than LO3 and day) can be seen in JAS compared to
AMJ in all of these regions. In general, the dominant meteorological drivers from the
observation-based MLRs in these internal regions are RH and Tx. The contribution of
RH is evident in AMJ (e.g. ME, or EA), while Tx is clearly dominant in JAS. SSRD is
also a key driver of MDA8 O3 and generally, the wind factors (W10m and Wdir)
appear to have a minor contribution.
Despite the CTM-based MLRs being able to capture the meteorological predictors, we
observe discrepancies among the internal regions when compared to the observation-
based MLR. The inter-model differences in terms of the relative importance of
predictors are greater in AMJ than in JAS. For instance, the contribution of the LO3 is
overestimated by most of CTMs, specifically WRF-Chem that shows a larger sensitivity
to LO3 in both seasons over all of these regions. Similarly, EMEP also shows a larger
contribution of LO3 than the rest of the CTMs, particularly in AMJ. Substantial
differences are found in the influence of RH when comparing the observation-based and
the CTMs-based models. The CTMs do not capture the relative importance of the RH
well, especially in AMJ. In general, the CTMs driven by WRF meteorology show a
slightly larger contribution of RH in most of the cases, although we notice that there are
also some differences among the models that share the same meteorology. CTMs do
capture the relative importance of Tx in all regions, but overall they overestimate it, as





they also show for SSRD. Here, we find discrepancies when comparing the contribution of predictors in the statistical models from CTMs driven by the same meteorology (e.g. EMEP and WRF-Chem when compared to CHIMERE and MINNI). The largest differences among the CTMs are found for WRF-Chem, which tends to underestimate the contribution of the meteorological drivers in most of the regions. Interestingly, as mentions in Section 2, this is the only online coupled model participating in EDT.

Figure 7 presents the relative importance of individual predictors in the MLRs developed at the external regions (Fig. 1) for both seasons. The observation-based MLRs show that the main driving factor is LO3 in AMJ, while the effect of meteorological drivers becomes stronger in JAS. RH presents a larger contribution in some regions (e.g. IN, IP or SC) in AMJ and Tx in JAS (e.g. IN, IP, SC and BA). The contribution of wind components, Wdir and W10m, is mainly reflected in both seasons in the western regions (i.e. IN and IP) and in MD, respectively.

Overall, all CTMs show this tendency, although there are substantial differences when comparing the individual drivers' contribution in the observation-based and CTM-based MLRs, particularly in AMJ (Fig. 7). CTMs do not capture the contribution of LO3 reflected by the observation-based MLRs. As in the previous analysis (section 4.1) the largest discrepancies are found in BA, where observation-based MLR shows that most of the variability of ozone would be explained by LO3. On the contrary the CTM-based MLRs underestimate the contribution of LO3 and overestimate the meteorological effect in terms of larger contribution of Tx, SSRD and RH (e.g. LOTOS, CHIMERE and MINNI). The contribution of RH is underestimated by the CTMs in most of the regions, (except in BA). On the contrary, the relative importance of SSRD is overestimated in some regions (e.g. IP, IN or MD) and Tx (IN, SC), in particular for the CTMs driven by WRF. Overall, CTMs show the observed contribution of W10m and Wdir in both seasons, although with some inconsistences among the regions and CTMs.

Our results indicate that the relative importance of meteorological factors is stronger in the internal regions (Fig.6) than in the external regions (Fig.7), which could be partially attributed to a larger variability of most of the meteorological fields in internal regions (Fig. S1). The external regions are also more likely to be influenced by the lateral boundary conditions applied by each CTM. In addition, in some external regions (e.g. IP or MD), as mentioned in section 2, the use of a coarser grid in some regions might be insufficient to capture mesoscale processes, such as land-sea breezes, which also control MDA8 O3 concentrations (Millán et al. 2002). Moreover, we observe that meteorology becomes more important in summer, when local photochemistry processes are dominant. In general, CTMs show this tendency, but limitations to reproduce the effect of some meteorological drivers are found. Specifically, while CTMs tend to overestimate the contribution of Tx, and SSRD, they underestimate the relative importance of RH, which is also reflected in the correlations coefficients between predictand the predictors (Figs. S2, S3).

4.3.2 Sensitivity of ozone to the drivers

We assess the sensitivities of MDA8 O3 to the drivers through their standardised coefficients obtained in the MLR (Section 3). These coefficients provide further information about the changes of MDA8 O3 due to effect of each driver. Figures 8 and 9 depict the values of the main driving factors obtained in the MLR for the internal and



the external regions (respectively): LO3, Tx and RH. Similarly to those patterns described by the relative importance of drivers, we observe that the ozone response to LO3 is stronger in AMJ than in JAS: the corresponding standardised coefficients are always positive and generally higher in AMJ. The observed sensitivities to LO3 are smaller in the internal regions (Fig. 8), being particularly dominant in the external regions (Fig. 9). Overall, most of the CTMs reflect a similar tendency. However, there are evident differences among observations and CTMs when comparing the values of the standardised coefficients, specifically in some regions such as BA or MD. When comparing the ozone responses of the CTMs to LO3, we observe that in most of the regions MATCH and MINNI show values closest to observations, while WRF-Chem shows a large sensitivity to LO3.

Correlations between MDA8 O3 and Tx are strong, especially in the internal regions in JAS (Fig. S2). Overall, we show that the CTMs appear to capture the observed effect of Tx better in JAS than in AMJ in most of the regions. The highest sensitivities to Tx are found in some internal regions such as ME, NI, FR and EN, which is also shown in the CTMs. However, we see that most of the CTMs tend to overestimate the effect of Tx. Moreover, distinct sensitivities to Tx are shown by models that share the same meteorology (i.e. CHIMERE, EMEP, MINNI and WRF-Chem). In particular, the MINNI and CHIMERE models show higher Tx sensitivities when compared to the rest of the CTMs. While MINNI model presents the highest sensitivities to Tx in spring, specifically in EN and FR, EMEP shows smaller values and it underestimates the correlations between Tx and MDA8 O3 (Figs. S2, S3).

The slope of the ozone-temperature relationship ($m_{O3-T}$) has been used in several studies to assess the ozone climate penalty (eg. Bloomer et al., 2009, Steiner et al., 2010, Rasmussen et al., 2012, Brown-Steiner et al. 2015) in the context of future air quality. Thus, we additionally analyse the relationship ozone-temperature in order to provide insight into the ability of CTMs to reproduce the observed $m_{O3-T}$. Similarly as in previous work (Brown-Steiner et al. 2015), the slopes are obtained from a simple linear regression using only Tx (without the influence from other predictors) and they are used to quantify such relationship in both seasons, AMJ and JAS.

Figures 10 and 11 illustrate the $m_{O3-T}$ for the internal and the external regions respectively. The observed $m_{O3-T}$ is larger in JAS than in AMJ. In AMJ, it ranges between -0.45 and 1.15 ppbK$^{-1}$ with the largest values found in ME, NI and MD. In JAS, the observed climate penalty is of the order of 1-2.7 ppbK$^{-1}$ with the largest values in EN, FR, ME, NI, and MD. CTMs show a better agreement with observations in JAS than in AMJ. CTMs tend to overestimate the climate penalty in AMJ in most of the regions, with some exceptions, such as EMEP and MATCH that systematically underestimate the slopes. Also, CTMs are generally better in simulating the observed $m_{O3-T}$ in the internal regions compared to the $m_{O3-T}$ in the external regions, where in general CTMs appear to overestimate the climate penalty in both seasons. Using this metric, we identify some regions particularly sensitive to temperature, with larger values of $m_{O3-T}$ (e.g. EN, ME, FR, NI or MD). Through a multi-model assessment, Colette et al. (2015) showed a significant summertime climate penalty in southern, western and central European regions (e.g. EA, IP, FR, ME or MD) in the majority of the future climate scenarios used. Our study shows that most of the CTMs confirm the observed climate penalty in JAS in such regions in the near present, although we found



that most of the CTMs overestimate the climate penalty in AMJ, especially in the
external regions.
We see a stronger effect of RH in AMJ than in JAS in the observations compared with
the CTMs (Figs. 8 and 9), with the greatest impact in the internal regions (e.g. EA, ME,
NI, FR and EN). The CTMs show this tendency slightly in some regions (e.g. ME, FR
or EN), but differences become evident when compared to the observed values and
overall they underestimate the effect of RH. As mentioned, CTMs underestimate the
strength of the relationship between ozone-RH (Figs. S2, S3). This general lack of
sensitivity to RH could also partially explain the tendency for all CTMs to show a high
bias in simulated ozone compared with observations (Fig. 2).  Among the possible
reasons for this inconsistency, we hypothesize that it can be related to the fact that
ozone removal processes can be associated to higher relative humidity levels during
thunderstorm activity on hot moist days, which might not be well captured by CTMs.
Furthermore, the documented impacts of ozone dry deposition suggest that it may also
play a role in explaining the problems that CTMs show to reproduce the observed
relationship ozone-relative humidity.
High SSRD levels favour photochemical ozone formation and it is usually positively
correlated to ozone. In this case, CTMs also present some limitations to capture this
effect and they overestimated the sensitivities of ozone to SSRD (Figs. S4, S5). For
example, the observations show lower and surprisingly negative effect of SSRD.
Although the correlations between SSRD and ozone are positive (see Fig. S2, S3), the
presence of other predictors in the regression may reverse the sign of the estimated
coefficient. The CTMs show a stronger sensitivity of ozone to SSRD and they
overestimate its influence on surface ozone. Similarly, the sensitivities to Wdir and
W10m are also overestimated by the CTMs, especially in AMJ (Figs. S4, S5).
Our analysis suggests that CTMs present more limitations to reproduce the influence of
meteorological drivers to MDA8 O3 concentrations in the external regions than in the
internal regions, particularly in AMJ. Moreover, we find the largest discrepancies in
BA, where models show the poorest seasonal performance and correlation coefficients
(Figs. 2 and 3, respectively), probably due a low quality of the observational dataset.
Furthermore, LO3 is the main driver over most of the external regions and explains a
large proportion to the total variability of MDA8 O3, while meteorological factors play
a smaller influence. Lemaire et al. (2016) found a very low performance (based on $R^2$)
over the British Isles, Scandinavia and the Mediterranean using a different statistical
approach that only included two meteorological drivers. They attributed this low skill to
the large influence over those regions of long-range transport of air pollution (Lemaire
et al. 2016). Our results confirm the small influence of the meteorological drivers over
those regions and the strong influence of the ozone persistence. Moreover, in the case of
the external regions of northern Europe, it could also be explained due to the dominance
of transport processes such as the stratospheric-tropospheric exchange or long-range
transport from the European continent, rather than local meteorology, particularly in
AMJ (Monks, 2000, Tang et al. 2009, Andersson et al. 2009).
Previous work pointed out that local sources of NOx and biogenic VOC (ozone
precursors) are important factors of summertime ozone pollution in the Mediterranean
basin (Richards et al. 2013). Moreover, some studies suggested that the local vertical



recirculation and accumulation of pollutants play an important role in ozone pollution
episodes in this region: during the nighttime the air masses are held offshore by land-sea
breeze, creating reservoirs of pollutants that are brought the following day (Millán et al.
20002, Jiménez et al. 2006, Querol et al. 2017). All of these factors (e.g. local emissions
as well as local and large-scale processes) control the ozone variability, which might
explain the smaller influence of local meteorological factors shown in this study over
the Mediterranean basin when compared to meteorological influence in the internal
regions. Thus, we may hypothesize that the strong impact of LO3 observed in the
external regions over southern Europe (i.e. IP, MD, BA) could be partially due to the
role of vertical accumulation and recirculation of air masses along the Mediterranean
coasts as a result of the mesoscale phenomena, which is enhanced by the complex
terrains that surround the Basin. Other important factor for the strong impact of LO3
observed is the slow dry deposition of ozone on water that would favour the ozone
persistence in southern Europe.
Overall we conclude that CTMs capture the effect of meteorological drivers better in the
internal regions (EN, FR, ME, NI and EA), where the influence of local meteorological
conditions is stronger. The major effect of meteorological parameters found in the
internal European regions might be also attributed to the fact that overall the variability
of meteorological conditions is larger in those regions (Fig. S1). We also find
differences among the CTMs driven by the same meteorology. As mentioned in the
introduction, Bessagnet el al. (2016) suggested that the spread in the model results
could partly explained by the differences in the vertical diffusion coefficient and the
planetary boundary layer, differently diagnosed in each of the CTMs. Our results also
indicate that even though models share the same meteorology (considering the
prescribed requirements defined by the EDT exercise) they show discrepancies when
compared to each other, which could be attributed other sources of uncertainties (such
as physical and chemical internal process in the CTMs). The NMVOC and $NO_x$
emissions from the biosphere are critical in the ozone formation. Since biogenic
emissions were not specifically prescribed, which have a strong dependence on
temperature and solar radiation, discrepancies in the CTMs performances, (e.g. different
sensitivities to Tx) might be expected. Furthermore, we notice that the CTMs do not
reproduce consistently the regional ozone-temperature relationship, which is a key
factor when assessing the impacts of climate change on future air quality.
**5. Summary and conclusions**
The present study evaluates the capability of a set of Chemical Transport Models
(CTMs) to represent the regional relationship between daily maximum 8-hour average
ozone (MDA8 O3) and meteorology over Europe. Our results show systematic
differences between the CTMs in reproducing the seasonal cycle when compared to
observations. In general, they tend to overestimate the MDA8 O3 in most of the
regions. In the western and northern regions (i.e. Inflow, England and Scandinavia),
some models did not capture the high ozone levels in spring (e.g. CHIMERE, MINNI
and WRF-Chem), while in other southern regions (e.g. Iberian Peninsula,
Mediterranean and Balkans) they overestimated the ozone levels in summer (e.g.
LOTOS, CHIMERE). Of the CTMs, MATCH and MINNI were the most successful in
capturing the observed seasonal cycle of ozone in most regions. All CTMs revealed
limitations to reproduce the variability of ozone over the Balkans region, with a general
overestimation of the ozone concentrations, considerably larger during the warmer



months (July, August). As reflected in the results, a limitation of the interpolated
observational product used here is that in some regions (e.g. southern Europe) it has a
lower quality due to a reduced number of stations (section 2.1).
The MLRs performed similarly for most of the CTMs and observations, describing
more than 60 % of the total variance of MDA8 O3. Overall, the MLRs perform better in
JAS than in AMJ, and the highest percentages of described variance were found in Mid
Europe and North Italy. This could be attributed to local photochemical processes being
more important in JAS, and is consistent with a stronger influence of long-range
transport in AMJ.
The effects of predictors revealed spatial and seasonal patterns, in terms of their relative
importance in the MLRs. Particularly, we noticed a larger local meteorological
influence in the regions located towards the interior, here termed as the internal regions
(i.e. England, France, Mid-Europe, North Italy and East-Europe). A minor local
meteorological contribution was found in the rest of the regions, referred as the external
regions (i.e. Inflow, Iberian Peninsula, Scandinavia, Mediterranean and Balkans). The
CTMs are in better agreement with the observations in the internal regions than in the
external regions, where they were not as successful in reproducing the effects of the
ozone drivers. Overall, the different behaviour in the MLRs developed in the external
regions could be attributed to (i) a larger influence of dynamical processes rather than
local meteorological processes (e.g. long range transport in the northern regions) (ii) a
stronger impact of the boundary conditions (iii) the use of a coarser grid that might be
insufficient to capture mesoscale processes that also influence MDA8 O3 (e.g. sea-land
breezes in the southern regions).
We found substantial differences in the sensitivities of MDA8 O3 to the different
meteorological factors among the CTMs, even when they used the same meteorology.
As Bessagnet et al. (2016) point out, the differences amongst CTMs could be partly
attributed to some other diagnosed model variables (e.g. vertical diffusion coefficient
and boundary layer height, as well as vertical model resolution). To assess the effect of
such potential sources of uncertainties, further investigations would be required.
Moreover, variations in the sensitivity of ozone to meteorological parameters could
depend on differences in the chemical and photolysis mechanisms and the
implementation of various physics schemes, all of which differ between the CTMs (see
Colette et al. 2017a). Specifically, the discrepancies found in the sensitivities of MDA8
O3 to maximum temperature might be also attributed to biogenic emissions not
prescribed in the models. This was particularly reflected in the analysis of the slopes
ozone-temperature ($m_{O3-T}$) to assess the climate penalty, which differed between CTMs
and regions when compared to the observations in both seasons. Most of the CTMs
confirm the observed climate penalty in JAS, but with larger discrepancies in the
external regions than in the internal regions. Furthermore, CTMs tend to overestimate
the climate penalty in AMJ (particularly in the external regions).
Our results have shown that CTMs tend to overestimate the influence of maximum
temperature and surface solar radiation in most of the regions, both strongly associated
with ozone production. None of the CTMs captured the strength of the observed
relationship between ozone and relative humidity appropriately, underestimating the
effect of relative humidity, a key factor in the ozone removal processes. We speculate
that ozone dry deposition schemes used by the CTMs in this study may not adequately



represent the relationship between humidity and stomatal conductance, thus
underestimating the ozone sink due to stomatal uptake. Further sensitivity analyses
would be recommended for testing the impact of the current dry deposition schemes in
the CTMs.
**Data availability**
The data are available upon request from the corresponding author.
**Acknowledgments**

We acknowledge Jordan L. Schnell for providing the interpolated dataset of MDA 8 O3.
Modelling data used in the present analysis were produced in the framework of the
EURODELTA-Trends Project initiated by the Task Force on Measurement and
Modelling of the Convention on Long Range Transboundary Air Pollution.
EURODELTA-Trends is coordinated by INERIS and involves modelling teams of
BSC, CEREA, CIEMAT, ENEA, IASS, JRC, MET Norway, TNO, SMHI. The views
expressed in this study are those of the authors and do not necessarily represent the
views of EURODELTA-Trends modelling teams.





**List of Tables:**

| CTM | Meteorology | Coupling |
|---|---|---|
| LOTOS-EUROS | RACMO2 | Off-line |
| MATCH | HIRLAM | Off-line |
| EMEP | WRF | Off-line |
| CHIMERE | | Off-line |
| MINNI | | Off-line |
| WRF-Chem | | On-line |

**Table 1**. List of the chemistry-transport models used in the study, their corresponding meteorological
driver and chemistry/meteorology coupling.

| Predictor | Definition | |
|---|---|---|
| LO3 | Lag of O3 (24 h) | 853 |
| Tx | Maximum temperature | 854 |
| RH | Relative humidity | 855 |
| SSRD | Surface solar radiation | 856 |
| Wdir | Wind direction | 857 |
| W10m | Wind speed | 858 |
| day | $\sin(2\pi d_t/365.25)$, $\cos(2\pi d_t/365.25)$ | 859 860 861 862 |

**Table 2.** List of the predictors used in the multiple linear regression analysis: meteorological parameters,
lag of O3 (24h, previous day) and the seasonal cycle components.

| Region | Acronym | Coordinates (longitude, latitude) | |
|---|---|---|---|
| England | EN | 5W-2E, 50N-55N | 867 868 |
| Inflow | IN | 10W-5W, 50N-60N, and 5W-2E, 55N-60N | 869 870 |
| Iberian Peninsula | IP | 10W-3E, 36N-44N | 871 |
| France | FR | 5W-5E, 44N-50N | 872 |
| Mid-Europe | ME | 2E-16E, 48N-55N | 873 |
| Scandinavia | SC | 5E-16E, 55N-70N | 874 875 |
| North Italy | NI | 5E-16E, 44N-48N | 876 |
| Balkans | BA | 18E-28E, 38N-44N | 877 |
| Mediterranean | MD | 3E-18E, 36N-44N | 878 879 |
| Eastern Europe | EA | 16E-30E, 44N-55N | 880 881 |

**Table 3.** List of the regions with the short name and the coordinates.



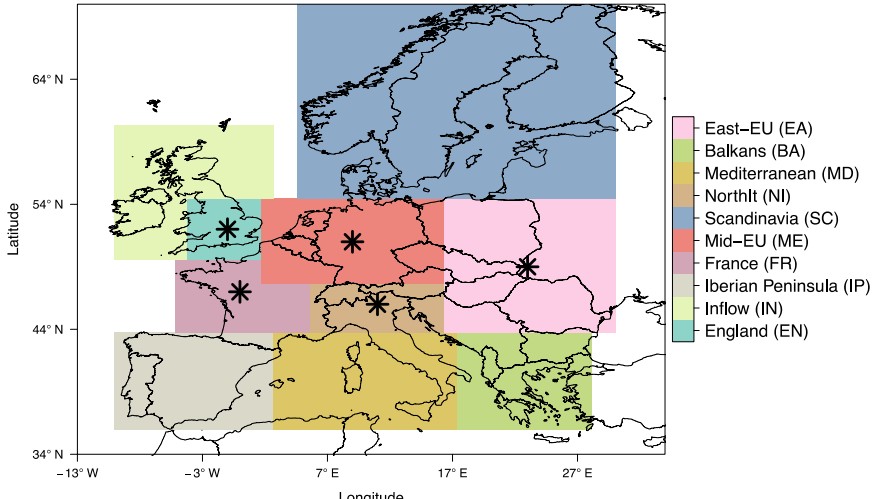

**Figure 1.** Map of the regions considered in the study. Regions indicated with a black star are referred to
the internal regions in the text. The rest of regions are referred to the external regions of the European
domain.

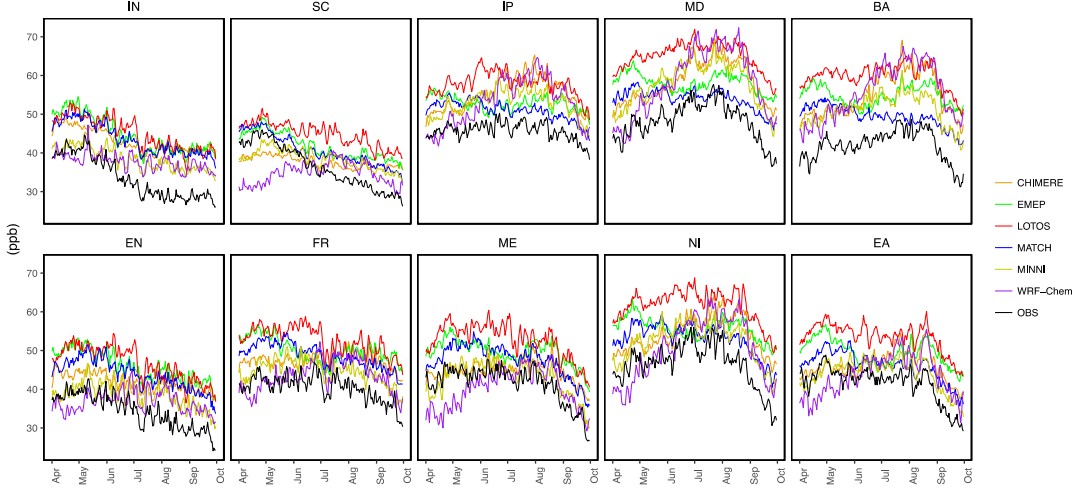

**Figure 2.** Time series of daily averages of MDA8 O3 during the ozone season (April-September) for the
period of study (2000-2010) at each subregion.





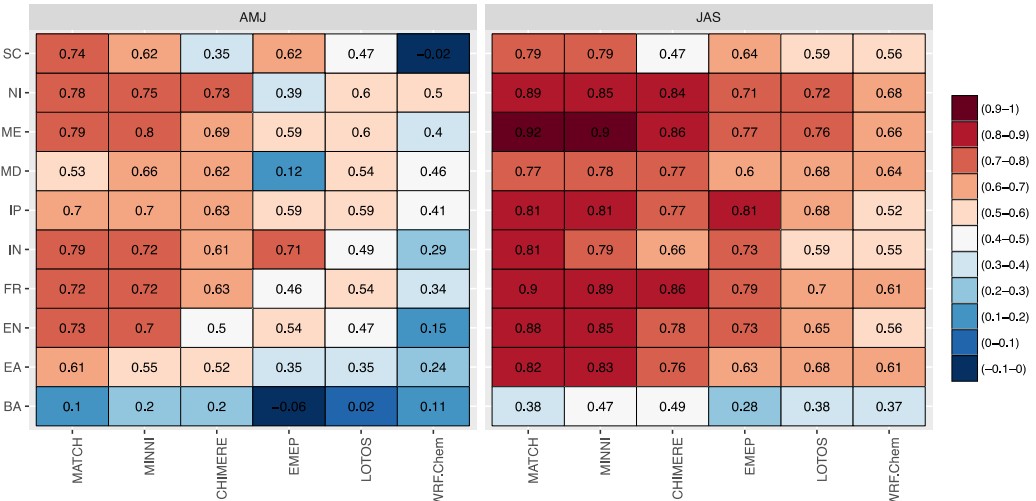

**Figure 3.** Correlation coefficients between observed and modelled MDA8 O3 for spring (AMJ) and
summer (JAS) for the period of study (2000-2010) at each region (rows) and models (columns, ordered
by highest correlation values).

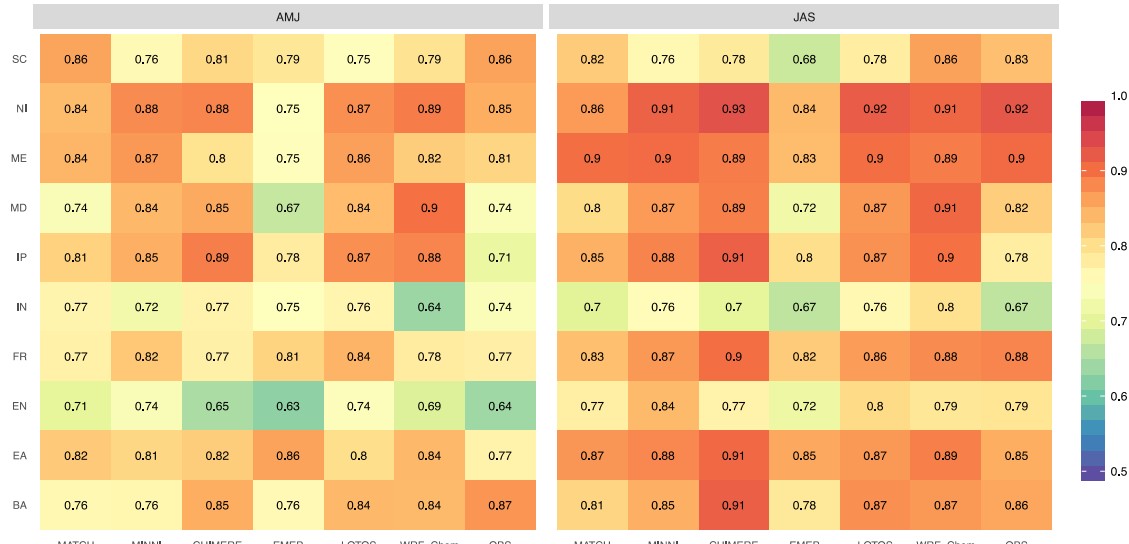

**Figure 4.** Coefficients of determination ($R^2$) for each CTM-based (ordered as in Fig.3) and observation-
based MLR in spring (AMJ) and summer (JAS).






| | MATCH | MINNI | CHIMERE | EMEP | LOTOS | WRF–Chem | OBS |
|---|---|---|---|---|---|---|---|
| SC | 1.61 | 2.04 | 1.3 | 1.85 | 2.39 | 2.23 | 2.02 |
| NI | 2.35 | 2.76 | 2.31 | 2.34 | 2.38 | 3.46 | 3.23 |
| ME | 2.21 | 2.96 | 2.48 | 2.29 | 2.65 | 3.53 | 3.2 |
| MD | 1.93 | 3.15 | 2.27 | 2.27 | 2.14 | 3.17 | 3.74 |
| IP | 1.64 | 2.56 | 1.96 | 2.01 | 2.18 | 2.87 | 2.85 |
| IN | 2.17 | 2.81 | 1.8 | 2.83 | 3.04 | 3.51 | 2.81 |
| FR | 2.26 | 2.98 | 2.23 | 2.3 | 2.79 | 3.7 | 3.29 |
| EN | 2.78 | 3.69 | 2.69 | 3.19 | 3.33 | 4.06 | 3.8 |
| EA | 1.84 | 2.36 | 1.95 | 1.76 | 2.36 | 2.85 | 3.12 |
| BA | 1.75 | 2.54 | 2.14 | 1.88 | 1.92 | 2.93 | 3.32 |

AMJ

| | MATCH | MINNI | CHIMERE | EMEP | LOTOS | WRF–Chem | OBS |
|---|---|---|---|---|---|---|---|
| SC | 1.61 | 2.01 | 1.44 | 1.9 | 2.65 | 2.19 | 1.95 |
| NI | 2.58 | 2.8 | 2.77 | 2.58 | 2.66 | 3.91 | 3.06 |
| ME | 2.21 | 2.94 | 2.96 | 2.69 | 3.09 | 3.62 | 3.3 |
| MD | 2.01 | 3.4 | 2.79 | 2.64 | 2.64 | 3.5 | 3.79 |
| IP | 1.54 | 2.49 | 2.27 | 2.21 | 2.3 | 3.21 | 2.9 |
| IN | 2.3 | 2.78 | 2.04 | 3.04 | 3.17 | 3.05 | 2.84 |
| FR | 2.35 | 3.21 | 2.77 | 3.06 | 3.3 | 3.81 | 3.27 |
| EN | 2.86 | 3.48 | 2.84 | 3.44 | 3.59 | 4.14 | 3.66 |
| EA | 1.88 | 2.44 | 2.69 | 2.12 | 2.74 | 3.11 | 3.14 |
| BA | 1.72 | 2.84 | 2.59 | 2.21 | 2.35 | 3.32 | 3.63 |

JAS

**Figure 5.** Root mean square errors (RMSE) for each CTM-based (ordered as in Fig.3) and observation-
based MLR at each region, in spring (AMJ) and summer (JAS).

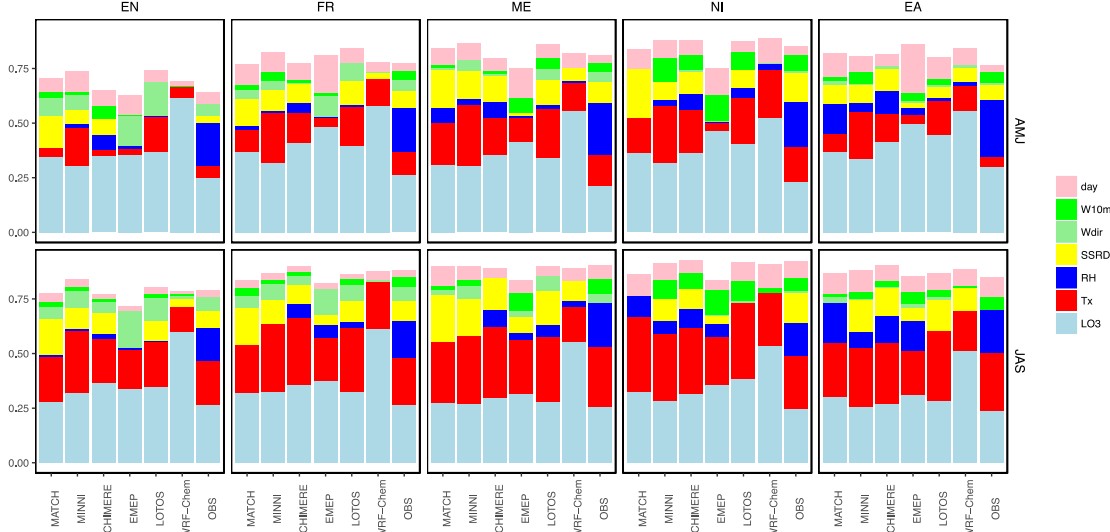

**Figure 6.** Proportion of each predictor to the total explained variance for each CTM-based (ordered as in
Fig.3) and observation-based MLR in AMJ (top) and JAS (bottom) for the internal regions: England
(EN), France (FR), Mid-Europe (ME), North Italy (NI) and East-Europe (EA).





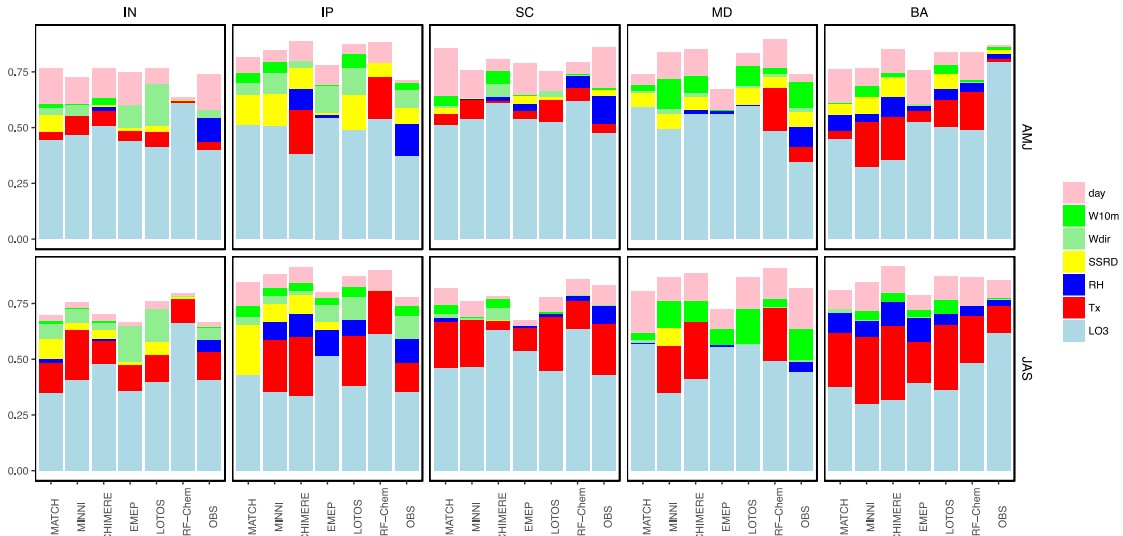

**Figure 7.** Proportion of each predictor to the total explained variance for each CTM-based (ordered as in
Fig.3) and observation-based MLR in AMJ (top) and JAS (bottom) for the external regions: Inflow (IN),
Iberian Peninsula (IP), Scandinavia (SC), Mediterranean (ME) and Balkans (BA).

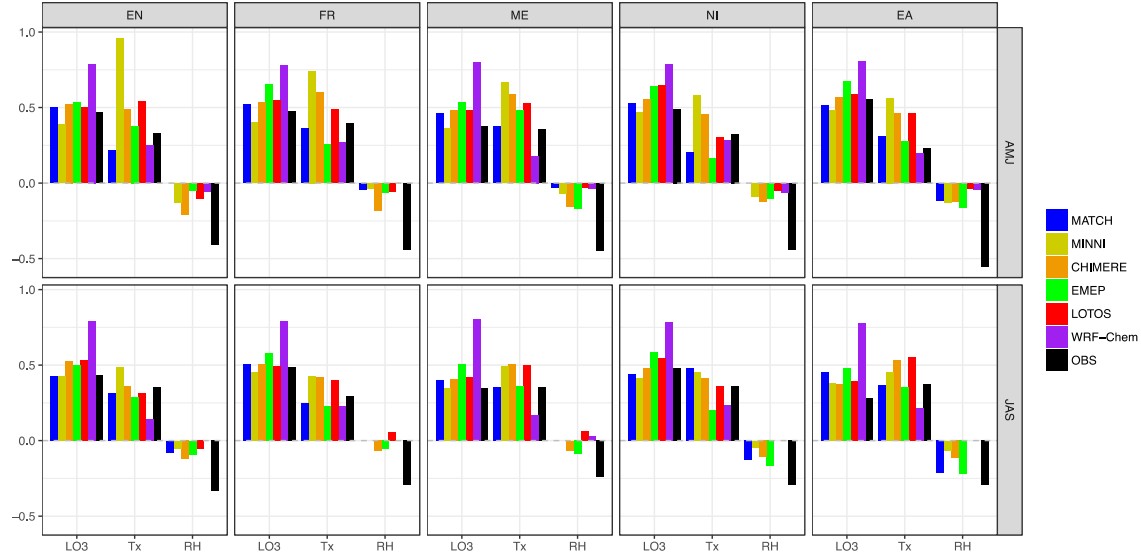

**Figure 8.** Standardised coefficients values of the main key-driving factors (LO3, Tx and RH) for each
CTM-based (ordered as in Fig.3) and observation-based MLR in AMJ (top) and JAS (bottom) and for the
internal regions: England (EN), France (FR), Mid-Europe (ME), North Italy (NI) and East-Europe (EA).






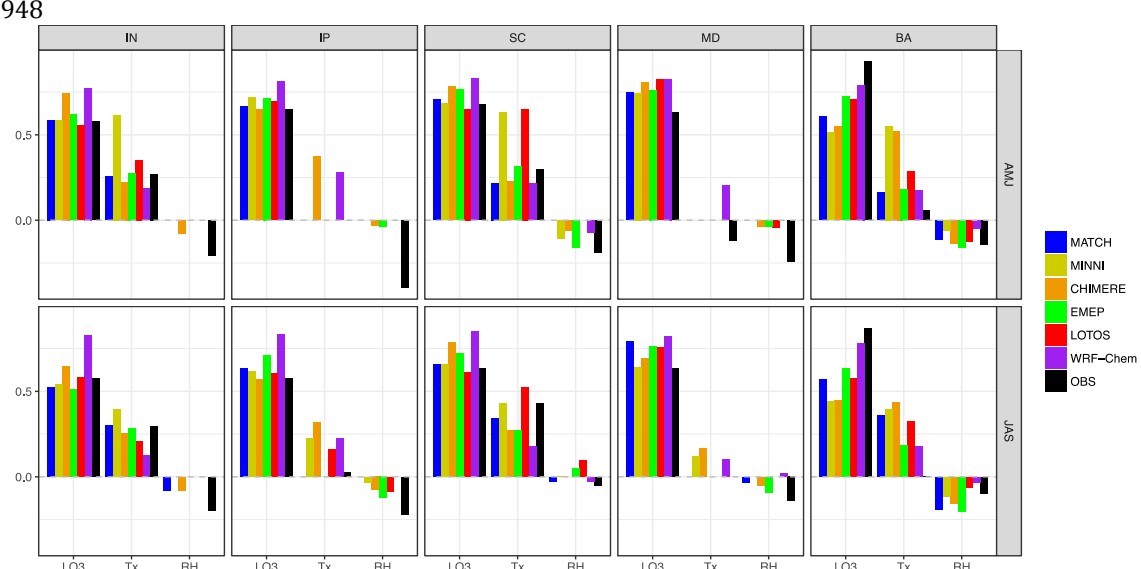

**Figure 9.** Standardised coefficients values of the main key-driving factors (LO3, Tx and RH) for each
CTM-based (ordered as in Fig.3) and observation-based MLR in AMJ (top) and JAS (bottom) and for the
external regions: Inflow (IN), Iberian Peninsula (IP), Scandinavia (SC), Mediterranean (ME) and Balkans
(BA).

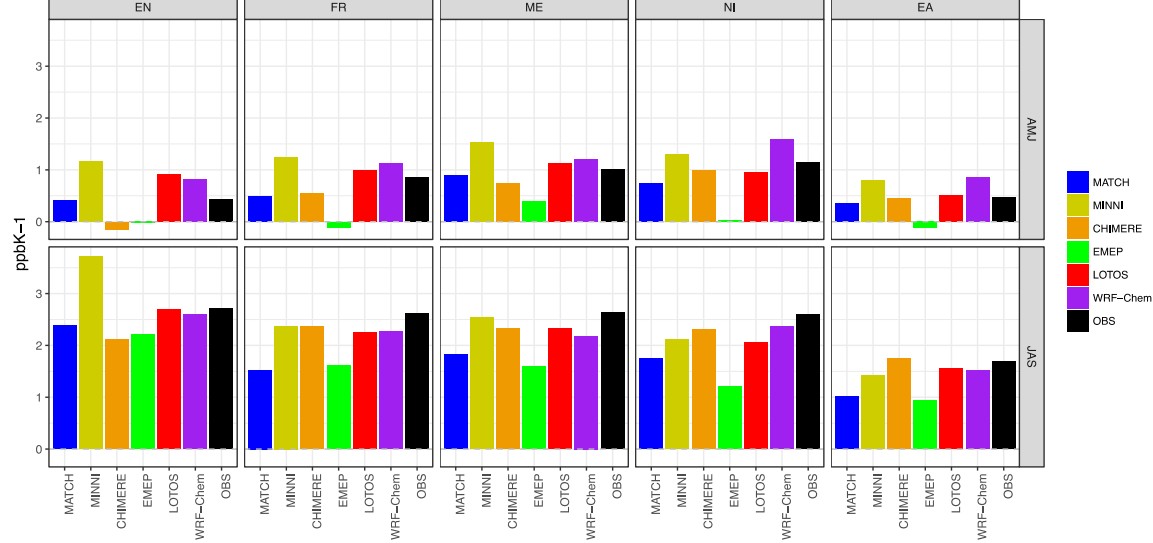

**Figure 10.** Slopes ($m_{O3-T}$; $ppbK^{-1}$) obtained from a simple linear regression to estimate the relationship
ozone-temperature for each CTM-based (ordered as in Fig.3) and observation-based MLR in AMJ (top)
and JAS (bottom) and for the internal regions: England (EN), France (FR), Mid-EU (ME), North Italy
(NI), East-EU (EA).





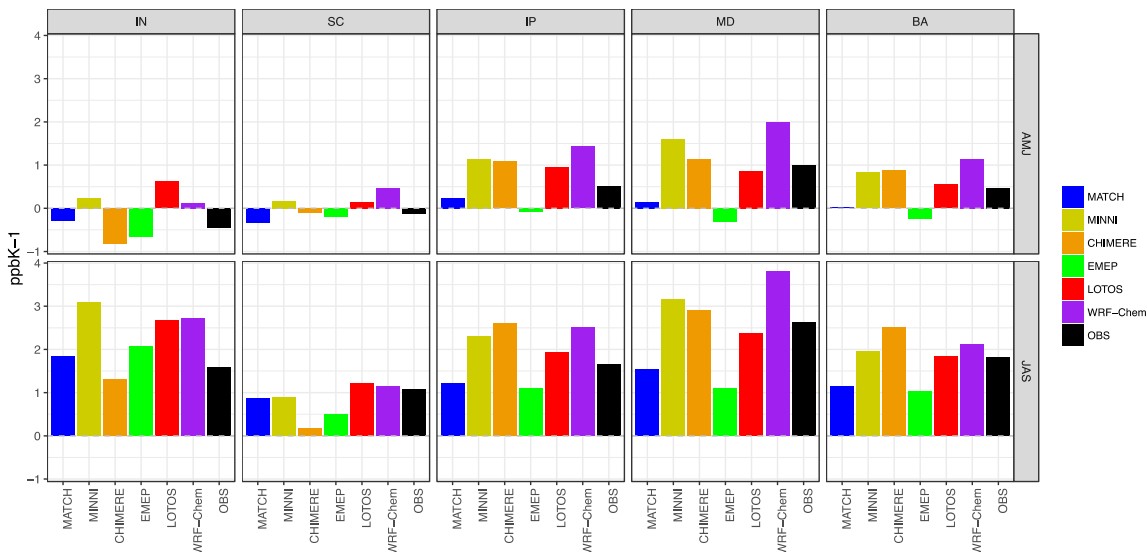


**Figure 11**. Slopes ($m_{O3-T}$; ppbK$^{-1}$) obtained from a simple linear regression to estimate the relationship
ozone-temperature for each CTM-based (ordered as in Fig.3) and observation-based MLR in AMJ (top)
and JAS (bottom) and for the external regions: Inflow (IN), Iberian Peninsula (IP), Scandinavia (SC),
Mediterranean (ME) and Balkans (BA).




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
