# Peer review of "A multi-model comparison of meteorological drivers of surface ozone 1"

_Atmospheric Chemistry and Physics, 2017_

## Referee Comment (RC1) · Anonymous Referee #3 · 15 May 2018

GENERAL REMARKS

A strong connection exists between air quality - in particular the surface ozone concentration - and accompanying meteorological conditions; hot, sunny, stable conditions favour formation of ozone while turbulent and cloudy condtions are assocoated with low ozone concentrations. It is crucial that air quality models correctly represent this connection. This paper uses data from observations and a number of state-of-science air quality models to investigate this issue. Using simple multiple linear regression models the relationship between ozone and a number of key meteorological quantities is analysed and compared to analogous MLR based on observations. The authors find that model performance varies with variable and geographic region analysed. Model performance with respect to temperature is commonly good while more limited perfor-

mance is found in case of the ozone-relative humidity relationship for all models. It is concluded that model resolution, boundary conditions and the parameterization of ozone dry deposition can have an important impact on model performance in addition to the meteorological variables under investigation.

This paper addresses an important question in air quality modelling. Obviously, if models fail to adequately represent the relationship that exists between meteorology and atmospheric chemistry/composition then air quality assessments and mitigation strategies based on those will be flawed. The use of multi-model datasets and the relatively simple approache of MLR seem appropriate and serve their purpose. Tables and Figures are used appropriately throughout the text and support well the findings. The conclusions drawn at the end are somewhat sparse and limited but interesting and useful. The only real discrepancy that exists in this study is the definition of the spring and summer seasons. I understand that shifting these back by one month may have benefits with respect to the ozone chemistry in the models but springtime is springtime for a good reason. The meteorological variability in springtime (March, April, May) has a profound impact on atmospheric chemistry and so has the comparative stability with its hot, dry and sunny conditions in summer (in general). I am not sure it is such a good idea to give up on the definition of the seasons but I am not going to make this a make-or-break condition for the paper to be published because I can foresee and endless discussion on the pros and cons with potentially little impact on the study at hand.

In my opinion the paper represents an important contribution to understanding model performance and potential discrepancies. However, I do not consider it a scientific milestone. Overall, I have read this manuscript with interest. If some minor issues and typping errors are corrected I believe the paper can be published.

SPECIFIC COMMENTS

L214: something is missing in this sentence after "observed"; please correct.

L217: insert "on" after "meteorological influence".

L271: should read "... defined from a climatology of observational data ..." or "... defined from climatologies of observational data ..."

L272: "including" instead of "included"

L305: "emission densities" instead of "emissions densities"

L352: nothing wrong but IMO the sentence would read more easily this way: "the domains covered by observations and CTMs do not coincide exactly"

L438: better: "... shows ozone peak concentrations in ..." or "... in the EMEP model ozone concentrations peak in April while ..."

L467: "products" instead of "product"

L552: "mentioned" instead of "mentions"

L569-571: it appears to me that these two sentences contradict each other; please clarify.

L617: insert "the" after "While"

L658: "associated with" instead of "associated to"

L667: insert "a" after "show" to read "... observasions show a lower ..."

L698: insert "back" after "brought" to read "... are brought back the following ..."

L718: insert "be" after "partly" to read "... could partly be explained ..."

L722: insert "to" after "attributed" to read "... could be attributed to other ..."

---

## Referee Comment (RC2) · Anonymous Referee #4 · 16 May 2018

Otero et al. evaluate the ability of a suite of state-of-the-art regional air quality models in their ability to reproduce the observed relationship between meteorological variables and surface ozone over Europe. They use a multiple linear regression approach, harnessing their previous experience using MLR from an observation-only standpoint. Their results are very relevant given the simulations for CMIP6 are beginning, providing context for future simulations based on how well models represent the ozone-meteorology relationship in present day. Although some of their results are mostly speculative, I understand that it is difficult to fully diagnose the potential issues within each model without further sensitivity simulations. I would also like to see a more quantitative discussion of the results, as the paper in its current form is much more qualitative. The paper is well written but requires minor revision prior to publication in

[Figure]

ACP.

General comments:

The introduction is quite long and contains information that is not highly relevant to the paper, which seems, at least in part, due to some excessive self-citation. I suggest trimming some of the more basic background details as well as some reorganization for clarity. For example, paragraphs 2-5 could be condensed into a single "how meteorology affects AQ" paragraph.

Although general numbers can be gleaned from the figures, the results are very lacking in the amount of quantitative statistics portrayed in the main text. Much of the discussion is very surface level and doesn't really provide the reader with new information other than the models are different from one another and observations. In the entire results section, there are only five or six specific numbers quoted. Obviously numbers shouldn't just be reported for the sake of reporting, but I'm certain the reader could benefit from more.

It would be helpful if the authors could put some of the results in the context of the differences between the models; i.e., "model A,B, and C may show X due to their representation of Y". This may require a bit of digging into the additional model diagnostics (e.g., biogenic emission rates) but it would help to provide some additional insight.

Specific comments:

Page 2, lines 84-85: referred *TO* as *THE* climate penalty

Page 2, line 89: remove "the", *concluded that climate change*

Page 3, lines 116-117: "meteorological dependence..." this sentence is oddly worded

Page 3, lines 118-119: Here and elsewhere, when describing metorological relationships with ozone, it should be worded "the ozone–relative humidity relationship" or "the relationship between ozone and relative humidity"

Page 3, lines 126-127: The statement about wind speed is abrupt and out of place.

Page 4, lines 186-187: Which studies?

Page 4, line 196: Mid-Atlantic U.S. states?

Page 5, line 217: influence *ON* MDA8 O3

Page 5, line 217 and elsewhere: Curious, but why not use subscripts for O3?

Table 1: This could be expanded so the reader does not have to refer elsewhere.

Page 8, line 415: *MDA8*

Page 9, lines 433-435: The sentence: "Models show discrepancies..." is extremely vague.

Page 10, line 485: the phrase "that, in general show lower values of R2 in JAS than in AMJ" is unnecessary since "with the exception" implies the reverse of what was said previously.

Page 10, line 486: No need to say "certain regions such as", just say which ones.

Page 11, line 518: Please elaborate on "non-local processes".

Page 12, line 595: due to *THE* effect

Page 14, lines 656-659: Please elaborate, are you saying that thunderstorms are not represented well? What part of the meteorological-chemistry relationship?

Page 14, lines 660-662: Sentence is awkwardly worded, also provide a reference if you say "the documented" anything.

Page 15, line 723: *processes*

---

## Referee Comment (RC3) · Anonymous Referee #2 · 18 May 2018

The paper presents an analysis of a suite of chemical transport models applied to simulate ozone levels over Europe. I value very high the community effort of gathering around a common exercise and I am aware of the amount of time required to run, collect, harmonise and analyse model's results. I have to note, however, that the comments I posted for the 'quick' assessment of the paper have not been addressed:

'The methodology presented is sound and entails massive amount of analysis work. Though, I fail to see the fruits of such analysis! I found the paper unbalanced in several parts, with a very long and qualitative introduction and scarcity of quantitative results. What is the main message of the paper? What the advancement? What is novel (in the results) with respect to other existing multi model comparison activities? Again, the authors conclude with speculation as to why certain models behave in some way

rather than in another?. But where are the quantitative, supporting argument? I would invite the authors to deeply analyse existing results from the literature (e.g. Vautard et al., 2012 Brunner et al, 2015; Makar et al. 2015, among many others) and reconsider their contribution in light of the novelty it might bring.'

After re-reading the paper I still fail to see what is the key novelty and scientific advancement brought by this paper. All findings (sensitivity to season, regions, etc..) are documented by dozens of papers. The authors apply a, perhaps, different methodology that converges, nonetheless, at conclusions similar to those already known since several years. In my view the paper, in its current shape, is too qualitative and lacks a clear message that stands out and justifies a publication. In light of the poorly exposed scientific significance I advise the editor to ask the authors to significantly review the paper before it can be considered suitable for publication.

---

## Author Comment (AC1) · 18 May 2018

Since the initial submission of the manuscript (August 2017), the ongoing EURODELTA Trend exercise has delivered new data provided by specific working groups. Therefore, we have updated the data for the model MATCH, which has fixed a bug in emissions and improved its wet scavenging parameterization. In addition, further analyses of the WRFChem runs revealed inconsistencies in the model step-up affecting other variables and consequently the surface ozone performance, resulting in the exclusion of those runs from the EURODELTA exercise. Therefore, we have decided to also exclude the WRF-Chem model from our analysis. Neither of these two changes significantly influences the results and the final message presented in this study. However, to reflect this update, minor modifications to the manuscript and the supplementary material will

be required in a small number of places. The corresponding line numbers as well as the revised figures are detailed below. Line numbers refer to the current online discussion version of the manuscript.

Lines 137-142, 144-146 will be removed.

Line 265 will be changed

Lines 282,283 will be removed

Lines 292-297, 298-301 will be removed

Lines 430,431 will be changed

Lines 441,442 will be removed

Lines 469,472 will be removed

Lines 473,474 will be changed

Line 491 will be changed

Line 503 will be changed

Lines 538-540 will be removed

Lines 549-552 will be removed

Lines 606,607 will be removed

Line 615 will be modified

Line 740 will be changed

Please also note the supplement to this comment:
https://www.atmos-chem-phys-discuss.net/acp-2017-787/acp-2017-787-AC1-supplement.pdf

[Figure]

[Figure]

**Fig. 1.** Time series of daily averages of MDA8 O3during the ozone season (April-September) for the period of study (2000-2010) at each subregion.

[Figure]

**Fig. 2.** Correlation coefficients between observed and modelled MDA8 O3for spring (AMJ) and summer (JAS) for the period of study (2000-2010) at each region (rows) and models (columns, ordered by highest cor.)

[Figure]

|  | AMJ | | | | | | JAS | | | | | |
|---|---|---|---|---|---|---|---|---|---|---|---|---|
|  | MATCH | MINNI | CHIMERE | EMEP | LOTOS | OBS | MATCH | MINNI | CHIMERE | EMEP | LOTOS | OBS |
| SC | 0.86 | 0.76 | 0.81 | 0.79 | 0.75 | 0.86 | 0.84 | 0.76 | 0.78 | 0.68 | 0.78 | 0.83 |
| NI | 0.83 | 0.88 | 0.88 | 0.75 | 0.87 | 0.85 | 0.91 | 0.91 | 0.93 | 0.84 | 0.92 | 0.92 |
| ME | 0.83 | 0.87 | 0.8 | 0.75 | 0.86 | 0.81 | 0.9 | 0.9 | 0.89 | 0.83 | 0.9 | 0.9 |
| MD | 0.73 | 0.84 | 0.85 | 0.67 | 0.84 | 0.74 | 0.81 | 0.87 | 0.89 | 0.72 | 0.87 | 0.82 |
| IP | 0.81 | 0.85 | 0.89 | 0.78 | 0.87 | 0.71 | 0.86 | 0.88 | 0.91 | 0.8 | 0.87 | 0.78 |
| IN | 0.77 | 0.72 | 0.77 | 0.75 | 0.76 | 0.74 | 0.69 | 0.76 | 0.7 | 0.67 | 0.76 | 0.67 |
| FR | 0.75 | 0.82 | 0.77 | 0.81 | 0.84 | 0.77 | 0.84 | 0.87 | 0.9 | 0.82 | 0.86 | 0.88 |
| EN | 0.69 | 0.74 | 0.65 | 0.63 | 0.74 | 0.64 | 0.78 | 0.84 | 0.77 | 0.72 | 0.8 | 0.79 |
| EA | 0.82 | 0.81 | 0.82 | 0.86 | 0.8 | 0.77 | 0.89 | 0.88 | 0.91 | 0.85 | 0.87 | 0.85 |
| BA | 0.74 | 0.76 | 0.85 | 0.76 | 0.84 | 0.87 | 0.84 | 0.85 | 0.91 | 0.78 | 0.87 | 0.86 |

**Fig. 3.** Coefficients of determination (R2) for each CTM-based (ordered as in Fig.3) and observation-based MLR in spring (AMJ) and summer (JAS).

[Figure]

|  | AMJ | | | | | | JAS | | | | | |
|---|---|---|---|---|---|---|---|---|---|---|---|---|
| **SC** | 1.4 | 2.04 | 1.3 | 1.85 | 2.39 | 2.02 | 1.41 | 2.01 | 1.44 | 1.9 | 2.65 | 1.95 |
| **NI** | 2.14 | 2.76 | 2.31 | 2.34 | 2.38 | 3.23 | 2.04 | 2.8 | 2.77 | 2.58 | 2.66 | 3.06 |
| **ME** | 2.02 | 2.96 | 2.48 | 2.29 | 2.65 | 3.2 | 2 | 2.94 | 2.96 | 2.69 | 3.09 | 3.3 |
| **MD** | 1.78 | 3.15 | 2.27 | 2.27 | 2.14 | 3.74 | 1.91 | 3.4 | 2.79 | 2.64 | 2.64 | 3.79 |
| **IP** | 1.5 | 2.56 | 1.96 | 2.01 | 2.18 | 2.85 | 1.41 | 2.49 | 2.27 | 2.21 | 2.3 | 2.9 |
| **IN** | 1.91 | 2.81 | 1.8 | 2.83 | 3.04 | 2.81 | 2.06 | 2.78 | 2.04 | 3.04 | 3.17 | 2.84 |
| **FR** | 2.05 | 2.98 | 2.23 | 2.3 | 2.79 | 3.29 | 2.12 | 3.21 | 2.77 | 3.06 | 3.3 | 3.27 |
| **EN** | 2.59 | 3.69 | 2.69 | 3.19 | 3.33 | 3.8 | 2.62 | 3.48 | 2.84 | 3.44 | 3.59 | 3.66 |
| **EA** | 1.65 | 2.36 | 1.95 | 1.76 | 2.36 | 3.12 | 1.66 | 2.44 | 2.69 | 2.12 | 2.74 | 3.14 |
| **BA** | 1.66 | 2.54 | 2.14 | 1.88 | 1.92 | 3.32 | 1.56 | 2.84 | 2.59 | 2.21 | 2.35 | 3.63 |
|  | MATCH | MINNI | CHIMERE | EMEP | LOTOS | OBS | MATCH | MINNI | CHIMERE | EMEP | LOTOS | OBS |

**Fig. 4.** Root mean square errors (RMSE) for each CTM-based (ordered as in Fig.3) and observation-based MLR at each region, in spring (AMJ) and summer (JAS).

[Figure]

**Fig. 5.** Proportion of each predictor to the total explained variance for each CTM-based (ordered as in Fig.3) and observation-based MLR in AMJ (top) and JAS (bottom) for the internal regions.

[Figure]

**Fig. 6.** Proportion of each predictor to the total explained variance for each CTM-based (ordered as in Fig.3) and observation-based MLR in AMJ (top) and JAS (bottom) for the external regions.

[Figure]

**Fig. 7.** Standardised coefficients values of the main key-driving factors (LO3, Tx and RH) for each CTMbased (ordered as in Fig.3) and observation-based MLR in AMJ and JAS for the internal regions.

[Figure]

**Fig. 8.** Standardised coefficients values of the main key-driving factors (LO3, Tx and RH) for each CTMbased (ordered as in Fig.3) and observation-based MLR in AMJ and JAS and for the external regions.

[Figure]

**Fig. 9.** Slopes (mO3-T; ppbK-1) obtained from a simple linear regression to estimate the relationship ozonetemperature for each CTM-based and observation-based MLR in AMJ and JAS for the internal regions.

[Figure]

**Fig. 10.** Slopes (mO3-T; ppbK-1) obtained from a simple linear regression to estimate the relationship ozonetemperature for each CTM-based and observation-based MLR in AMJ and JAS for the external regions

**Supplement:**

**Authors' comment on "A multi-model comparison of meteorological drivers of surface ozone over Europe" (acp-2017-787)**

**List of figures to be updated in the supplementary material**

[Figure]

**Figure S2**. Correlation coefficients between MDA8 O3 and each potential predictor used in the MLR. Correlations are computed for each season, AMJ (top) and JAS (bottom), and for internal regions: England (EN), France (FR), Mid-EU (ME), NI (North Italy), EA (East-EU).

[Figure]

**Figure S3**. Correlation coefficients between MDA8 O3 and each potential predictor used in the MLR. Correlations are computed for each season, AMJ (top) and JAS (bottom), and for external regions: IN (Inflow), SC (Scandinavia), IP (Iberian Peninsula), MD (Mediterranean) and Balkans (BA).

[Figure]

**Figure S4**. Standardised coefficients values of the rest of the meteorological predictors (SSRD, Wdir and W10m) for each CTM-based and observation-based MLR in AMJ (top) and JAS (bottom) and for the internal regions: England (EN), France (FR), Mid-Europe (ME), North Italy (NI) and East-Europe (EA).

[Figure]

**Figure S5**. Standardised coefficients values of the rest of the meteorological predictors (SSRD, Wdir and W10m) for each CTM-based and observation-based MLR in AMJ (top) and JAS (bottom) and for the external regions: Inflow (IN), Iberian Peninsula (IP), Scandinavia (SC), Mediterranean (ME) and Balkans (BA).

---

## Referee Comment (RC4) · Anonymous Referee #1 · 21 May 2018

The authors discuss an important question in air quality modeling, i.e. the capability of model results in reproducing several features of ozone time series. To this end, they compare observations with several state-of-the-art models. I think this issue is important and worthy of discussion, but I also think that the paper needs major revisions before it can be accepted for publication. First of all, I suggest to better balance the length of all sections; the introduction is too long but fails to highlight what has already been previously achieved and what are the main advances of this paper. What is really new with this work?

A second major concern is about the use of observations. As I can guess, the authors interpolate observations over a regular grid, but this introduces an additional problem. What is the representativeness of area-averaged observations? Usually, the support

of point observations is much more limited than 1°x1° grid cells. How the authors address this issue? Why not use a much simpler approach consisting of the comparison between observations and interpolated model values? As I can guess, Airbase observations contain several different station types (e.g. remote, suburban, urban, etc.). How are they treated?

A third comment concern the use of multiple models. The authors use a suite of model values, but they do not refer to any ensemble. I'm curious to know if an ensemble treatment may help in this case.

Finally, I also suggest a deeper analysis of the regression model. The authors implicitly assume a homoscedastic behavior. Is this supported by data? Due to the large interval of values and intrinsic periodicities in time series, I think that the variance cannot be assumed independent on model values. I suggest investigating on data properties, as well as on independence between the beta's values between different models and areas.

Typos: line 23: "ENE" should be "ENEA" line 161: "van Lon" should be "van Loon"

---

## Short Comment (SC1) · 27 May 2018

In different parts of this manuscript the authors mention that the largest discrepancies between modelled and observed MDA8 O3 are found for the Balkans. They attribute that to the low number of stations interpolated into the 1x1 degree grid cells in the dataset by Schell et al. (2014). I agree that is problem for the Balkans and for other "external regions", as correctly indicated by the authors, but our experience also shows that there are some inhomogeneities in that ozone dataset over the Balkans.

First, in Ordóñez et al. (2017) we examined the impact of high-latitude and subtropical anticyclones on surface ozone. That work found (i) upward ozone trends in that dataset over the Balkans and (ii) did not establish a clear impact of anticyclonic systems on ozone over the same region. Consequently, we omitted the Balkans from our regional analyses. Later on, Carro-Calvo et al. (2017) carried out a much more detailed evaluation of the quality of the ozone dataset before analysing the synoptic drivers of summer ozone in Europe. Figure S1 in the supplement of that paper displays the regions with some inhomogeneities prior to 2004. There is a small region with inhomogeneities over Scandinavia which should hardly affect your results and a much larger area covering most of your Balkan region. In Carro-Calvo et al. (2014) we decided to remove all O3 data over those regions before 2004.

Note that both Ordóñez et al. (2017) and Carro-Calvo et al. (2017) used a longer ozone dataset created by Jordan Schnell for a 15-year period (1998-2012). However, I have had a quick look at the shorter dataset used here and still see very low ozone mixing ratios in the Balkans during the first years. That might at least partly explain the high model biases (Figure 2) and low correlations (Figure 3) reported by this manuscript for that region. That could also have important implications for the results of the multiple linear regression models. As an example, the observation-based models suggest a very strong impact of ozone persistence in the Balkans, while that impact is not so strong for modelled ozone (Figure 7).

I would recommend the authors to plot the full time series of MDA8 O3 (daily values) averaged over that region and see if there is any break-point (I guess that around 2004) with a clear shift in the data. Then I would remove the data before that break-point and repeat all the analyses for that region.

In the last paragraph of page 14, the authors speculate on the reasons for the relatively low skill of the models in northern Europe: "Moreover, in the case of the external regions of northern Europe, it could also be explained due to the dominance of transport processes such as the stratospheric-tropospheric exchange or long-range transport from the European continent, rather than local meteorology, particularly in AMJ (Monks, 2000, Tang et al. 2009, Andersson et al. 2009)". According to the results of Carro-Calvo et al. (2017), I believe that is the case not only for spring but also for the

summer months (JJA in that paper).

Finally, I have read the manuscript with interest. I understand the reviewers' concerns but still think that some of the findings will be relevant for the community. As pointed out by one of the reviewers, "it is difficult to fully diagnose the potential issues within each model without further sensitivity simulations". The analysis of the results for the ensemble mean/median, as suggested by another reviewer, will not be sufficient to understand all the reasons for those discrepancies. However, that could help summarise some of the results and identify the meteorological drivers and processes (e.g. relative humidity, dry deposition?) which should be investigated in more detail in the future, through (i) careful evaluation of model parameterisations and (ii) sensitive simulations. Having that in mind, I am confident this manuscript will be a good contribution to the field. Some of its findings will hopefully raise our awareness about some processes which need to be better investigated in air quality models.

Carlos Ordóñez, Universidad Complutense de Madrid, Spain

References

Carro-Calvo, L., C. Ordóñez, R. García-Herrera, J. L. Schnell: Spatial clustering and meteorological drivers of summer ozone in Europe, Atmos. Environ., 167, 496-510,https://doi.org/10.1016/j.atmosenv.2017.08.050, 2017.

Ordóñez, C., Barriopedro, D., García-Herrera, R., Sousa, P. M., and Schnell, J. L.: Regional responses of surface ozone in Europe to the location of high-latitude blocks and subtropical ridges, Atmos. Chem. Phys., 17, 3111-3131, https://doi.org/10.5194/acp-17-3111-2017, 2017.

---

## Author Response (AR1)

**General author's comment, responses to referees and marked-up manuscript version**

**"A multi-model comparison of meteorological drivers of surface ozone over Europe" (acp-2017-787)**

Dear Editor and Referees,

We are pleased to see that the manuscript **"*A multi-model comparison of meteorological drivers of surface ozone over Europe*" (acp-2017-787)** has received great attention during the open discussion phase and we truly appreciate the constructive comments and suggestions from all the anonymous referees as well as the short comment and recommendations published by Dr. Carlos Ordóñez.

We are very happy to hear that in general the referees point out the valuable contribution of our manuscript to the air quality community. We also note the concerns from anonymous referee #2, who is less convinced regarding the novelty of the manuscript. In response to all comments, we have carefully revised the manuscript and have made major changes in the manuscript. Thanks to the referees' comments we consider that the paper is now improved, especially regarding the balance of the sections. We hope to have adequately addressed the referees' concerns and that they find the revised version of the manuscript suitable for publication.

In this letter we would like to provide a brief summary of the major changes in the revised version that have addressed similar issues, since some concerns were common between the four reviews of our manuscript. In the following, we list the main changes in the manuscript, which include:

- A substantial rewriting and reorganization of the introduction, removing redundant information and paragraphs not relevant for the paper.
- A deeper review of the existing literature to better clarify the key novelty of our study and highlight its main contribution to the air quality community.
- A more detailed discussion regarding our definition of the seasons. Accordingly new figures in the supplementary material are provided.
- A new detailed table with the main characteristics of the models is added.
- An additional time series analysis regarding the issue in the Balkans region is now added in the supplementary material (SC, Carlos Ordóñez).
- References and figures as well as the supplementary material have been accordingly modified.

We believe that all of these changes have lead to a significantly improved version of the paper, which in our opinion can certainly benefit the air quality modelling community.

Finally, we will be happy to clarify any question that might remain open and we would be glad to see the manuscript considered for publication.

Bellow, we provide a point-by-point response to the referees' comments (in ***bold italics***) following by our responses (in blue Font). Furthermore, a marked-up manuscript version showing the changes made is included.

**Author's response to Referee #1**

*Anonymous Referee #1-

**General comments**

*The authors discuss an important question in air quality modeling, i.e. the capability of model results in reproducing several features of ozone time series. To this end, they compare observations with several state-of-the-art models. I think this issue is important and worthy of discussion, but I also think that the paper needs major revisions before it can be accepted for publication. First of all, I suggest to better balance the length of all sections; the introduction is too long but fails to highlight what has already been previously achieved and what are the main advances of this paper. What is really new with this work?*

We are very grateful to the referee for the constructive comments and suggestions, which lead to improvement of this manuscript. We have carefully revised the manuscript and made the suggested changes.
The balance in some parts of manuscript, particularly the introduction, is a common concern from the referees. As stated in the general author's response, the revised version of the manuscript has significantly been improved according to the referees' suggestions.

*A second major concern is about the use of observations. As I can guess, the authors interpolate observations over a regular grid, but this introduces an additional problem. What is the representativeness of area-averaged observations? Usually, the support of point observations is much more limited than 1x1 grid cells. How the authors address this issue? Why not use a much simpler approach consisting of the comparison between observations and interpolated model values? As I can guess, Airbase observations contain several different station types (e.g. remote, suburban, urban, etc.). How are they treated?*

We would like to emphasise that we did not interpolate observations. As stated in the manuscript (see section 2.1) we have used a gridded dataset of MDA8 O3 provided by Schnell et al. (2014). This product was generated with objective mapping algorithm by merging thousands of stations from EMEP and Airbase. This dataset has been originally used by Schnell et al. (2014) to evaluate global air quality models and in Otero et al. (2016) to examine the influence of synoptic and meteorological conditions on MDA8 O3. Recently, Órdoñez et al. (2017) and Carro-Calvo et al. (2017) have been used this product to assess the impacts of high-latitude blocks and subtropical ridge on ozone and to examine a spatial clustering of meteorological drivers over Europe, respectively. Here, we have used this product for the first time to evaluate a set of regional CTMs over Europe.

*A third comment concern the use of multiple models. The authors use a suite of model values, but they do not refer to any ensemble. I'm curious to know if an ensemble treatment may help in this case.*

The CTMs included in this study did not use the same meteorological input: three of the models (CHIMERE, EMEP and MINNI) used WRF, while LOTOS was driven by

RACMO2 and MACTH by HIRLAM. Since we aim to identify potential deficiencies in the inputs of the CTMs, we consider that the use of an ensemble would not bring more insights to this study. Then, we decide not to include an ensemble mean.

*Finally, I also suggest a deeper analysis of the regression model. The authors implicitly assume a homoscedastic behaviour. Is this supported by data? Due to the large interval of values and intrinsic periodicities in time series, I think that the variance cannot be assumed independent on model values. I suggest investigating on data properties, as well as on independence between the beta's values between different models and areas.*

As referee#1 suggests, we have analysed the results from the statistical models (CTMs-based and observation-based MLR) during the model development. We follow a similar strategy to that of Otero et al. (2016), which included an analysis of multicollinearity between predictors and a further examination of the residuals. Figures 1 and 2 show the MLR diagnostics regarding the homoscedastic behaviour for both seasons, AMJ and JJA, respectively. In general, we did not observe from these plots a serious indication or pattern of heteroscedasticity. Additionally, we examined the distribution of the standardised residuals. Figures 3, 4 show the histograms of standardised residuals for both seasons AMJ and JJA, respectively. In addition, the normal probability plots (QQ-plots) of standardised residuals are depicted in figures 5 and 6. The diagnostic plots, did not reveal serious evidence that the residuals are not normally distributed. Therefore, we can conclude that generally, the MLRs were properly developed.

[Figure]

**Figure 1**. Standardized residuals versus predicted values for the observed-based and CTMs-based MRL in each region during springtime (AMJ).

[Figure]

**Figure 2**. Standardized residuals versus predicted values for the observed-based and CTMs-based MRL in each region during summertime (JAS).

[Figure]

[Figure]

**Figure 3**. Histograms of standardized residuals for the observed-based and CTMs-based MRL in each region during springtime (AMJ).

[Figure]

**Figure 4**. Histograms of standardized residuals for the observed-based and CTMs-based MRL in each region during summertime (JAS).

[Figure]

**Figure 5**. Normal probability plots (QQ-plots) of standardized residuals for the observed-based and CTMs-based MRL in each region during springtime (AMJ).

[Figure]

**Figure 6**. Normal probability plots (QQ-plots) of standardized residuals for the observed-based and CTMs-based MRL in each region during summertime (JAS).

***Typos: line 23: "ENE" should be "ENEA" line 161: "van Lon" should be "van Loon"***
Thank you. This has been corrected in the revised version.

We would like to thank referee #2 for the helpful and constructive comments to improve the quality of the manuscript. We have carefully studied the comments and suggestions and we have made a major revision of the original manuscript.  We hope that the revisions are acceptable and that our responses adequately address the comments.
Following the referee's comment, we have modified the current version of the manuscript in order to improve the quality and clarify the novelty and contribution of our study.

As the referee points out, in the recent literature several intercomparison and evaluation exercises of chemistry-transport models (CTMs) have contributed to better understand model uncertainties related the parameterization of processes and the quality of input data (e.g. Bessagnet et al. 2016, Solazzo et al. 2017, Vautard et al. 2012).  Previous intercomparison studies have evaluated the meteorological input data with different configurations used by air quality models (e.g. Vautard et al., 2012 and references therein). Most of these exercises were designed to cover short-time periods (e.g. one year, Vauvard et al. 2012, Brunner et al. 2015), while only a few numbers of studies included longer periods (e.g. Vautdard et al., 2006, Wilson et al., 2012), but only using one model. As stated in Colette et al., (2017), the design of EURODELTA-Trends (EDT) exercise serves to examine the long-term evolution of air quality and its drivers in Europe. In this context, the EDT experiment provided us a great opportunity to assess long-term air quality performance and the role of meteorological driving factors.

We understand the referee's concern regarding the novelty of this manuscript with respect to the previous cited studies, since we address a common objective by investigating the role of the meteorological input used by CTM. However, we believe that our study contributes to the existing multi-model comparison exercises in several unique ways:

- We have used an interpolated product (Schnell et al., 2015) that offers a good opportunity to directly compare model outputs, rather than using a reduced number of stations within a smaller spatial coverage.

- Our study presents a multi-model evaluation over the whole Europe analysing the model performance in 10 different subregions, while most of the cited studies focused on a smaller number of European regions.

- A common approach to evaluate the performance of the meteorological inputs that drive air quality models consists of comparing directly with observational datasets through standard statistical metrics (e.g. bias, correlations, RMSE, among others). However, our primary goal was not to evaluate the models' skill to reproduce observed parameters, but rather we aimed to provide an alternative method to examine whether the current air quality models are capable to reproduce the observed meteorological sensitivities of ozone that have been reported in a wide number of statistical modelling studies. Thus, systematic differences in the ozone response to the most important meteorological factors can provide a valuable diagnostic tool to identify potential deficiencies in the inputs and parameters of air quality models. Only a couple of studies have addressed this issue but only using one model (e.g. Davis et al. 2011, Fix et al. 2017). To our knowledge this is first study that presents a multi-model evaluation of ozone sensitivities to the main meteorological key drivers over Europe.

- One of the main outcomes from this study points out the limitations in the CTMs to reproduce observed relationships between ozone and some meteorological key drivers. In particular we found that CTMs underestimate the strength of ozone-relative humidity relationship, which might be related to the dry deposition schemes (as suggested in previous works, e.g. Kavassalis and Murphy, 2017). Due to the strict requirements of the EDT exercise in terms of input data, most of the differences in the model outputs can be attributed to the model formulation and set-up. Furthermore, it is beyond of the scope of this study to fully diagnose the issues within each model, which would require further sensitivity simulations and model set-up.

*After re-reading the paper I still fail to see what is the key novelty and scientific advancement brought by this paper. All findings (sensitivity to season, regions, etc..) are documented by dozens of papers. The authors apply a, perhaps, different methodology that converges, nonetheless, at conclusions similar to those already known since several years. In my view the paper, in its current shape, is too qualitative and lacks a clear message that stands out and justifies a publication. In light of the poorly exposed scientific significance I advise the editor to ask the authors to significantly review the paper before it can be considered suitable for publication.*

We appreciate the referee's comment and we agree that the manuscript needed to clarify the novelty and the contribution to the existing literature. We have carefully addressed these suggestions that are now reflected in the revised version of the manuscript. As the other referees pointed out, we do believe that our study represents a valuable contribution to the ACP community.

**References**

[revised manuscript text omitted]

**Author's response to Referee #3**

*Anonymous Referee #3

**GENERAL REMARKS**

*A strong connection exists between air quality - in particular the surface ozone concentration and accompanying meteorological conditions;  hot,  sunny,  stable conditions favour formation of ozone while turbulent and cloudy conditions are associated with low ozone concentrations.  It is crucial that air quality models correctly represent this connection. This paper uses data from observations and a number of state-of-science air quality models to investigate this issue.   Using simple multiple linear regression models the relationship between ozone and a number of key meteorological quantities is analysed and compared to analogous MLR based on observations. The authors find that model performance varies with variable and geographic region analysed.  Model performance with respect to temperature is commonly good while more limited performance is found in case of the ozone-relative humidity relationship for all models.  It is concluded that model resolution, boundary conditions and the parameterization of ozone dry deposition can have an important impact on model performance in addition to the meteorological variables under investigation. This paper addresses an important question in air quality modelling. Obviously, if models fail to adequately represent the relationship that exists between meteorology and atmospheric chemistry/composition then air quality assessments and mitigation strategies based on those will be flawed. The use of multi-model datasets and the relatively simple approach of MLR seem appropriate and serve their purpose. Tables and Figures are used appropriately throughout the text and support well the findings.   The conclusions drawn at the end are somewhat sparse and limited but interesting and useful. The only real discrepancy that exists in this study is the definition of the spring and summer seasons.  I understand that shifting these back by one month may have benefits with respect to the ozone chemistry in the models but springtime is springtime for a good reason. The meteorological variability in springtime (March, April, May) has a profound impact on atmospheric chemistry and so has the comparative stability with its hot, dry and sunny conditions in summer (in general).  I am not sure it is such a good idea to give up on the definition of the seasons but I am not going to make this a make-or-break condition for the paper to be published because I can foresee and endless discussion on the pros and cons with potentially little impact on the study at hand.*
*In my opinion the paper represents an important contribution to understanding model performance and potential discrepancies. However, I do not consider it a scientific milestone. Overall, I have read this manuscript with interest. If some minor issues and typing errors are corrected I believe the paper can be published.*

Firstly, we would like to thank the referee for acknowledging the contribution of the manuscript and the careful reading of the manuscript.

We understand the referee's concern regarding the use of two seasons that differ from the meteorological seasons. As stated in the manuscript, we have selected two three month-periods, namely April-May-June (AMJ) and July-August-September (JAS) in order to examine separately the role of the meteorological parameters during the early and late parts of the European "ozone season", which typically lasts from April to September.

Following the recommendation of Referee #3, we have performed a sensitivity test using the meteorological seasons (March-April-May, MAM and June-July-August, JJA) to address this concern. The main results obtained from this analysis are shown in Figs. 1-5, and summarised below. Furthermore, we have stressed the choice of the seasons in the revised version of the manuscript including Figs. 4 and 5 in the Supplement.

Figure 1 illustrates the seasonal cycle of daily averages of MDA8 O3 for each month over the whole period of study. As depicted in Figure 1, the high ozone mixing ratios are typically observed between April-September, beginning midway through the meteorological spring season (March-April-May), and extending beyond the meteorological summer season (June-July-August). Then, we consider that our choice of 3-month periods covering the whole ozone season is particularly interesting to examine the impact of individual meteorological parameters when ozone shows high levels.

[Figure]

Figure1. Time series of daily averages of MDA8 O3 during the all period of study (2000-2010) at each subregion.

Figures 2 and 3 show the relative importance of individual predictors in the MLRs developed at the internal and external regions for both meteorological seasons (MAM, JJA) respectively. As shown in the manuscript (see Figures 6 and 7) the influence of the meteorological factors is stronger in the internal regions than in the external regions. When comparing the individual drivers' contribution in each meteorological season (.g. top, MAM and bottom, JJA figures 2, 3) we observe relatively small differences in the influence of meteorological parameters on ozone variability between internal and external regions.

[Figure]

Figure2. Contribution of each predictor to the total explained variance for each CTM-based and observation for the internal regions: England (EN), France (FR), Mid-Europe (ME), North Italy (NI) and East-Europe (EA).

[Figure]

Figure3. Contribution of each predictor to the total explained variance for each CTM-based and observation for the external regions: Inflow (IN), Iberian Peninsula (IP), Scandinavia (SC), Mediterranean (ME) and Balkans (BA).

Similarly, as described in the manuscript we also found notable differences between the re-defined seasons (AMJ, JAS) in most of the regions. In particular in the case of maximum temperature (Tx) and relative humidity (RH), two key-driving factors, it can be shown the different effects, in terms of their individual contribution to the total explained variance, when using both season definitions (MAM, JAS and AMJ, JAS). Figures 4 and 5 show the values of the relative importance of Tx and RH respectively for the seasons AMJ, JAS (top of figures 4,5) and MAM, JJA (bottom of figures 3,4). In general, we notice a stronger influence of maximum temperature during the warmer months, referred to as JAS, than during the meteorological season, JJA (Figure 4). We also observe a larger contribution of relative humidity in AMJ in most of regions, especially in the internal regions (e.g. EA, FR, NI, EN) (figure 5). Therefore, our choice of the seasons allow us to better understand the different meteorological impacts on ozone variability when considering the early and late parts of the European ozone season (April-September). Furthermore, we consider that the contrast of drivers' contribution between seasons and regions is an interesting contribution of this study.

[Figure]

Figure 4. Values of relative importance (contribution to the total explained variance) of maximum temperature in the seasons AMJ, JAS and MAM and JJA.

[Figure]

Figure 5. Values of relative importance (contribution to the total explained variance) of relative humidity in the seasons AMJ, JAS and MAM and JJA.

**SPECIFIC COMMENTS**

*L214: something is missing in this sentence after "observed"; please correct.*
L214 has been changed.
*L217: insert "on" after "meteorological influence".*
L217 changed.
*L271: should read "... defined from a climatology of observational data ..." or "... defined from climatologies of observational data ..."*
L271 corrected.
*L272: "including" instead of "included"*
L272 corrected.
*L305: "emission densities" instead of "emissions densities"*
L305 corrected.
*L352: nothing wrong but IMO the sentence would read more easily this way: "the domains covered by observations and CTMs do not coincide exactly"*
Thanks for this suggestion. L352 has been modified.
*L438: better: "... shows ozone peak concentrations in ..." or "... in the EMEP model ozone concentrations peak in April while ..."*
L438 modified.
*L467: "products" instead of "product"*
L476 corrected.

*L552: "mentioned" instead of "mentions"*
L552 removed.

*L569-571: it appears to me that these two sentences contradict each other; please clarify.*
L569-571 modified.

*L617: insert "the" after "While"*
L617 modified

*L658: "associated with" instead of "associated to"*
L658 modified

*L667: insert "a" after "show" to read "... observasions show a lower ..."*
L667 modified

*L698: insert "back" after "brought" to read "... are brought back the following ..."*
L698 modified

*L718: insert "be" after "partly" to read "... could partly be explained ..."*
L718 modified

*L722: insert "to" after "attributed" to read "... could be attributed to other ..."*
L722 modified

**Author's response to Referee #4**

**Anonymous Referee #4**

**General comments:**

*Otero et al. evaluate the ability of a suite of state-of-the-art regional air quality models in their ability to reproduce the observed relationship between meteorological variables and surface ozone over Europe. They use a multiple linear regression approach, harnessing their previous experience using MLR from an observation-only standpoint. Their results are very relevant given the simulations for CMIP6 are beginning, providing context for future simulations based on how well models represent the ozone meteorology relationship in present day. Although some of their results are mostly speculative, I understand that it is difficult to fully diagnose the potential issues within each model without further sensitivity simulations. I would current form is much more qualitative. The paper is well written but requires minor revision prior to publication in ACP.*

We would like to thank the referee for the careful reading, and acknowledging the usefulness of the manuscript. We appreciate the insightful comments that have helped to improve the manuscript.

*The introduction is quite long and contains information that is not highly relevant to the paper, which seems, at least in part, due to some excessive self-citation. I suggest trimming some of the more basic background details as well as some reorganization for clarity. For example, paragraphs 2-5 could be condensed into a single "how meteorology affects AQ" paragraph. Although general numbers can be gleaned from the figures, the results are very lacking in the amount of quantitative statistics portrayed in the main text. Much of the discussion is very surface level and doesn't really provide the reader with new information other than the models are different from one another and observations. In the entire results section, there are only five or six specific numbers quoted. Obviously numbers shouldn't just be reported for the sake of reporting, but I'm certain the reader could benefit from more. It would be helpful if the authors could put some of the results in the context of the differences between the models; i.e., "model A,B, and C may show X due to their representation of Y". This may require a bit of digging into the additional model diagnostics (e.g., biogenic emission rates) but it would help to provide some additional insight.*

Thank you for your suggestions to improve the balance of the manuscript. This has been a common concern from others referees and as mentioned in the general author's response, the revised version of the manuscript has significantly been improved following the referees' suggestions.

We understand the referee's comment regarding the quantitative discussion. However we would like to highlight that in this case it is difficult to provide a full diagnostic about the specific issues of the CTMs without more simulations and sensitivity tests in the model set-up, which is beyond of the scope of this study. Considering the strict requirements of the input data in the EDT exercise, most of the discrepancies among the models found in this analysis can be attributed to the model formulation of the physical and chemical processes. Therefore, we believe that our study provides further insights and indications for future improvement and model development.

**Specific comments:**

*Page 2, lines 84-85: referred \*TO\* as \*THE\* climate penalty*
L84, 85 have been modified.

*Page 2, line 89: remove "the", \*concluded that climate change\**
L89 have been modified.

*Page 3, lines 116-117: "meteorological dependence: : :" this sentence is oddly worded*
L117 has been modified.

*Page 3, lines 118-119: Here and elsewhere, when describing meteorological relationships with ozone, it should be worded "the ozone–relative humidity relationship" or "the relationship between ozone and relative humidity"*
Thanks you for this comment. L118, 630, 660, 668 have been changed.

*Page 3, lines 126-127: The statement about wind speed is abrupt and out of place.*
Thanks you for this comment. L186-187 have been modified.

*Page 4, lines 186-187: Which studies?*
L186,187 have been modified.

*Page 4, line 196: Mid-Atlantic U.S. states?*
It is Eastern U.S. L196 has been slightly modified.

*Page 5, line 217: influence \*ON\* MDA8 O3*
L217 has been corrected.

*Page 5, line 217 and elsewhere: Curious, but why not use subscripts for O3?*
Thank for this comment. All subscript are now used in the text.

*Table 1: This could be expanded so the reader does not have to refer elsewhere.*
Thank for this comment. Table 1 has been modified.

*Page 8, line 415: \*MDA8\**
L415 has been corrected.

*Page 9, lines 433-435: The sentence: "Models show discrepancies: : :" is extremely vague.*
L433-435 have been modified.

*Page 10, line 485: the phrase "that, in general show lower values of R2 in JAS than in AMJ" is unnecessary since "with the exception" implies the reverse of what was said previously.*
L485 has been modified.

*Page 10, line 486: No need to say "certain regions such as", just say which ones.*
L486 has been corrected.

*Page 11, line 518: Please elaborate on "non-local processes".*
L518 has been modified.

*Page 12, line 595: due to \*THE\* effect*
L518 has been corrected.

*Page 14, lines 656-659: Please elaborate, are you saying that thunderstorms are not represented well? What part of the meteorological-chemistry relationship?*
Here, we are arguing that the contribution of the relative humidity in the MLR might represent the impact of afternoon thunderstorms or moister conditions that reduce high levels of ozone, which might be poorly captured by CTM-based MLRs (compared with the influence of relative humidity in the observed-based MLR).

***Page 14, lines 660-662: Sentence is awkwardly worded, also provide a reference if you say "the documented" anything.***
L660-662 have been modified.
***Page 15, line 723: \*processes\****
L723 has been corrected.

**Author's response to Short Comment from Dr.C. Ordóñez**

*In different parts of this manuscript the authors mention that the largest discrepancies between modelled and observed MDA8 O3 are found for the Balkans. They attribute that to the low number of stations interpolated into the 1x1 degree grid cells in the dataset by Schell et al. (2014). I agree that is problem for the Balkans and for other "external regions", as correctly indicated by the authors, but our experience also shows that there are some inhomogeneities in that ozone dataset over the Balkans.*

*First, in Ordóñez et al. (2017) we examined the impact of high-latitude and subtropical anticyclones on surface ozone. That work found (i) upward ozone trends in that dataset over the Balkans and (ii) did not establish a clear impact of anticyclonic systems on ozone over the same region. Consequently, we omitted the Balkans from our regional analyses. Later on, Carro-Calvo et al. (2017) carried out a much more detailed evaluation of the quality of the ozone dataset before analysing the synoptic drivers of summer ozone in Europe. Figure S1 in the supplement of that paper displays the regions with some inhomogeneities prior to 2004. There is a small region with inhomogeneities over Scandinavia which should hardly affect your results and a much larger area covering most of your Balkan region. In Carro-Calvo et al. (2014) we decided to remove all O3 data over those regions before 2004.Note that both Ordóñez et al. (2017) and Carro-Calvo et al. (2017) used a longer ozone dataset created by Jordan Schnell for a 15-year period (1998-2012). However, I have had a quick look at the shorter dataset used here and still see very low ozone mixing ratios in the Balkans during the first years. That might at least partly explain the high model biases (Figure 2) and low correlations (Figure 3) reported by this manuscript for that region. That could also have important implications for the results of the multiple linear regression models. As an example, the observation-based models suggest a very strong impact of ozone persistence in the Balkans, while that impact is not so strong for modelled ozone (Figure 7).*

*I would recommend the authors to plot the full time series of MDA8 O3 (daily values) averaged over that region and see if there is any break-point (I guess that around 2004) with a clear shift in the data. Then I would remove the data before that break-point and repeat all the analyses for that region.*

We would like to thank Dr. Carlos Ordóñez for the thorough revision of our manuscript and his useful comments.

We agree with you regarding the problem in the Balkans region. As we indicated in the manuscript, the evaluation in this region is particularly complicated due to the reduced number of stations. In addition, we think that the larger differences in this region (as showed in others so-called "*external*" regions) might be due to the impact of boundary conditions and the difficulty in capturing dynamical processes (such as recirculation patterns). Following your suggestion we have analysed the full time series of MDA 8 O₃ spatially averaged over each region. However, we did not observe a clear break point in the time series corresponding to the Balkans region (see revised version of the supplementary material). Therefore, we decided to include the whole dataset in our analysis. We agree that further analysis should investigate this issue in order to clarify and better understand the differences of this dataset over this particular region.

*In the last paragraph of page 14, the authors speculate on the reasons for the relatively low skill of the models in northern Europe: "Moreover, in the case of the external regions of northern Europe, it could also be explained due to the dominance of transport processes such as the stratospheric-tropospheric exchange or long-range transport from the European continent, rather than local meteorology, particularly in AMJ (Monks, 2000, Tang et al. 2009, Andersson et al. 2009)". According to the results of Carro-Calvo et al. (2017), I believe that is the case not only for spring but also for the summer months (JJA in that paper).*

Thank you for this comment. We agree that even in summer the meteorology has a smaller influence in the so-called *external* regions when compared with the meteorological impact in the *internal* regions (see Figs. 6 and 7 in the manuscript).

*Finally, I have read the manuscript with interest. I understand the reviewers' concerns but still think that some of the findings will be relevant for the community. As pointed out by one of the reviewers, "it is difficult to fully diagnose the potential issues within each model without further sensitivity simulations". The analysis of the results for the ensemble mean/median, as suggested by another reviewer, will not be sufficient to understand all the reasons for those discrepancies. However, that could help summarise some of the results and identify the meteorological drivers and processes (e.g. relative humidity, dry deposition?) which should be investigated in more detail in the future, through (i) careful evaluation of model parameterisations and (ii) sensitive simulations.*

We also understand the suggestion of the referee regarding the use of an ensemble mean. We would like to emphasize that our main goal is to assess the discrepancies between the CTMs using specific meteorological drivers. Therefore, as Carlos Ordóñez points out in this comment, we do not consider that the use of an ensemble mean can provide more insights within our evaluation.

*Having that in mind, I am confident this manuscript will be a good contribution to the field. Some of its findings will hopefully raise our awareness about some processes which need to be better investigated in air quality models.*

Finally, we appreciate the positive feedback acknowledging the contribution of our study.

[revised manuscript text omitted]
 **ve** 10 000m **sc:** First model level **dl:** 40m | MEGAN v2.04 (Guenther et al., 2006) | d**d:** Resistance model based on Wesely (1989) **sr:** Wesely (1989) | **lu:** Corine Land Cover 2006 (22 classes) **ad:** Blackman cubic polynomials (Yamartino,1993) **vd:** Kz approach following Lange (1989) |

| Predictor | Definition | 964 |
|---|---|---|
| LO3 | Lag of MDA8 $O_3$ (24 h) | 965 |
| Tx | Maximum temperature | 966 |
| RH | Relative humidity | 967 |
| SSRD | Surface solar radiation | 968 |
| Wdir | Wind direction | 969 |
| W10m | Wind speed | 970 |
| day | $\sin(2\pi d_t/365.25)$, $\cos(2\pi d_t/365.25)$ | 971 972 973 |

**Table 2.** List of the predictors used in the multiple linear regression analysis: meteorological parameters,
lag of MDA8 $O_3$  (24h, previous day) and the seasonal cycle components.

| Region | Acronym | Coordinates (longitude, latitude) | 978 |
|---|---|---|---|
| England | EN | 5W-2E, 50N-55N | 979 |
| Inflow | IN | 10W-5W, 50N-60N, and 5W-2E, 55N-60N | 980 981 |
| Iberian Peninsula | IP | 10W-3E, 36N-44N | 982 |
| France | FR | 5W-5E, 44N-50N | 983 |
| Mid-Europe | ME | 2E-16E, 48N-55N | 984 |
| Scandinavia | SC | 5E-16E, 55N-70N | 985 986 |
| North Italy | NI | 5E-16E, 44N-48N | 987 |
| Balkans | BA | 18E-28E, 38N-44N | 988 |
| Mediterranean | MD | 3E-18E, 36N-44N | 989 990 |
| Eastern Europe | EA | 16E-30E, 44N-55N | 991 |
|  |  |  | 992 |

**Table 3.** List of the regions with the short name and the coordinates.

[Figure]

**Figure 1.** Map of the regions considered in the study. Regions indicated with a black star are referred to
the internal regions in the text. The rest of regions are referred to the external regions of the European
domain.

**Comment [NF2]:** Figure 1 has been changed, only the legend has been modified (NorthIt, has been replaced by North Italy).

[Figure]

**Figure 2.** Time series of daily averages of MDA8 O₃O₃ during the ozone season (April-September) for
the period of study (2000-2010) at each subregion.

[Figure]

**Figure 3.** Correlation coefficients between observed and modelled MDA8 O₃O₃ for spring (AMJ) and
summer (JAS) for the period of study (2000-2010) at each region (rows) and models (columns, ordered
by highest correlation values).

[Figure]

**Figure 4.** Coefficients of determination ($R^2$) for each CTM-based (ordered as in Fig.3) and observation-
based MLR in spring (AMJ) and summer (JAS).

[Figure]

**Figure 5.** Root mean square errors (RMSE) for each CTM-based (ordered as in Fig.3) and observation-
based MLR at each region, in spring (AMJ) and summer (JAS).

[Figure]

**Figure 6.** Proportion of each predictor to the total explained variance for each CTM-based (ordered as in
Fig.3) and observation-based MLR in AMJ (top) and JAS (bottom) for the internal regions: England
(EN), France (FR), Mid-Europe (ME), North Italy (NI) and East-Europe (EA).

[Figure]

**Figure 7.** Proportion of each predictor to the total explained variance for each CTM-based (ordered as in
Fig.3) and observation-based MLR in AMJ (top) and JAS (bottom) for the external regions: Inflow (IN),
Iberian Peninsula (IP), Scandinavia (SC), Mediterranean (ME) and Balkans (BA).

[Figure]

**Figure 8.** Standardised coefficients values of the main key-driving factors (LO3, Tx and RH) for each
CTM-based (ordered as in Fig.3) and observation-based MLR in AMJ (top) and JAS (bottom) and for the
internal regions: England (EN), France (FR), Mid-Europe (ME), North Italy (NI) and East-Europe (EA).

[Figure]

**Figure 9.** Standardised coefficients values of the main key-driving factors (LO3, Tx and RH) for each
CTM-based (ordered as in Fig.3) and observation-based MLR in AMJ (top) and JAS (bottom) and for the
external regions: Inflow (IN), Iberian Peninsula (IP), Scandinavia (SC), Mediterranean (ME) and Balkans
(BA).

[Figure]

**Figure 10**. Slopes ($m_{O3-T}$; ppbK$^{-1}$) obtained from a simple linear regression to estimate the relationship
ozone-temperature for each CTM-based (ordered as in Fig.3) and observation-based MLR in AMJ (top)

and JAS (bottom) and for the internal regions: England (EN), France (FR), Mid-EU (ME), North Italy
(NI), East-EU (EA).

[Figure]

**Figure 11**. Slopes ($m_{O3-T}$; $ppbK^{-1}$) obtained from a simple linear regression to estimate the relationship
ozone-temperature for each CTM-based (ordered as in Fig.3) and observation-based MLR in AMJ (top)
and JAS (bottom) and for the external regions: Inflow (IN), Iberian Peninsula (IP), Scandinavia (SC),
Mediterranean (ME) and Balkans (BA).

**Comment [NF3]:** Figure 11 has been changed switching the order of SC and IP (wrong in the previous version).

[revised manuscript text omitted]